# Learning Discrete Concepts in Latent Hierarchical Models

**Lingjing Kong**[1], **Guangyi Chen**[1,2], **Biwei Huang**[3], **Eric P. Xing**[1,2], **Yuejie Chi**[1], and **Kun Zhang**[1,2]

[1]Carnegie Mellon University
[2]Mohamed bin Zayed University of Artificial Intelligence
[3]University of California San Diego

## Abstract

Learning concepts from natural high-dimensional data (e.g., images) holds potential in building human-aligned and interpretable machine learning models. Despite its encouraging prospect, formalization and theoretical insights into this crucial task are still lacking. In this work, we formalize concepts as discrete latent causal variables that are related via a hierarchical causal model that encodes different abstraction levels of concepts embedded in high-dimensional data (e.g., a dog breed and its eye shapes in natural images). We formulate conditions to facilitate the identification of the proposed causal model, which reveals when learning such concepts from unsupervised data is possible. Our conditions permit complex causal hierarchical structures beyond latent trees and multi-level directed acyclic graphs in prior work and can handle high-dimensional, continuous observed variables, which is well-suited for unstructured data modalities such as images. We substantiate our theoretical claims with synthetic data experiments. Further, we discuss our theory's implications for understanding the underlying mechanisms of latent diffusion models and provide corresponding empirical evidence for our theoretical insights.

## 1 Introduction

Learning semantic discrete concepts from unstructured high-dimensional data, such as images and text, is crucial to building machine learning models with interpretability, transferability, and compositionality, as empirically demonstrated by extensive existing work [1–11]. Despite these empirical successes, limited work is devoted to the theoretical front: the notions of concepts and their relations are often heuristically defined. For example, concept bottleneck models [12, 13] use human-specified annotations and recent methods [14–16] employ pretrained multimodal models like CLIP [17] to explain features with neural language. This lack of rigorous characterization impedes a deeper understanding of this task and the development of principled learning algorithms.

In natural images, the degree/extent of certain attributes (e.g., position, lighting) is often presented in a continuous form and main concepts of practical concern are often discrete in nature (e.g., object classes and shapes). Moreover, these concepts are often statistically dependent, with the dependence potentially resulting from some higher-level concepts. For example, the correlation between a specific dog's eye features and fur features may arise from a high-level concept for breeds (Figure 1). Similarly, even higher-level concepts may exist and induce dependence between high-level concepts, giving rise to a hierarchical model that characterizes all discrete concepts at different abstraction levels underlying high-dimensional data distributions. In this work, we focus on concepts that can be defined as discrete latent variables and related via a hierarchical model. Under this formalization, the query on the recoverability of concepts and their relations from unstructured high-dimensional distribution (e.g., images) amounts to the following causal identification problem:

38th Conference on Neural Information Processing Systems (NeurIPS 2024).

*Under what conditions is the discrete latent hierarchical causal model identifiable from high-dimensional continuous data distributions?*

Identification theory for latent hierarchical causal models has been a topic of sustained interest. Recent work [18–20] investigates identification conditions of latent hierarchical structures under the assumption that the latent variables are continuous and influence each other through linear functions. The linearity assumption fails to handle the general nonlinear influences among discrete variables. Another line of work focuses on discrete latent models. Pearl [21], Choi et al. [22] study latent trees with discrete observed variables. The tree structure can be over-simplified to capture the complex interactions among concepts from distinct abstract levels (e.g., multiple high-level concepts can jointly influence a lower-level one). Gu and Dunson [23] assume that binary latent variables can be exactly grouped into levels and causal edges often appear between adjacent levels, which can also be restrictive. Moreover, these papers assume observed variables are discrete, falling short of modeling the continuous distribution like images as the observed variables. Similar to our goal, Kivva et al. [24] show the discrete latent variables adjacent to the potentially continuous observed variables can be identified. However, their theory assumes the absence of higher-level latent variables and thus cannot handle latent hierarchical structures.

In this work, we show identification guarantees for the discrete hierarchical model under mild conditions on the generating function and causal structures. Specifically, we first show that when continuous observed variables (i.e., the leaves of the hierarchy) preserve the information of their adjacent discrete latent variables (i.e., direct parents in the graph), we can extract the discrete information from the continuous observations and further identify each discrete variable up to permutation indeterminacy. Given these "low-level" discrete latent variables, we establish graphical conditions to identify the discrete hierarchical model that fully explains the statistical dependence among the identified "low-level" discrete latent variables. Our conditions permit multiple paths within latent variable pairs and flexible locations of latent variables , encompassing a large family of graph structures including as special cases non-hierarchical structures [24], trees [21, 22, 25, 26] and multi-level directed acyclic graphs (DAGs) [23, 27] (see example graphs in Figure 2). Taken together, our work establishes theoretical results for identifying the discrete latent hierarchical model governing high-dimensional continuous observed variables, which to the best of our knowledge is the first effort in this direction. We corroborate our theoretical results with synthetic data experiments.

As an implication of our theorems, we discuss a novel interpretation of the state-of-the-art latent diffusion (LD) models [28] through the lens of a hierarchical concept model. We interpret the denoising objective at different noise levels as estimating latent concept embeddings at corresponding hierarchical levels in the causal model, where a higher noise level corresponds to high-level concepts. This perspective explains and unifies these seemingly orthogonal threads of empirical insights and gives rise to insights for potential empirical improvements. We deduce several insights from our theoretical results and verify them empirically. In summary, our main contributions are as follows.

- We formalize the framework of learning concepts from high-dimensional data as a latent-variable identification problem, capturing concepts at different abstraction levels and their interactions.

- We present identification theories for the discrete latent hierarchical model. To the best of our knowledge, our result is the first to address discrete latent hierarchical model beyond trees [21, 26] and multi-level DAGs [23] while capable of handling high-dimensional observed variables.

- We provide an interpretation of latent diffusion models as hierarchical concept learners. We supply empirical results to illustrate our interpretation and showcase its potential benefits in practice.

## 2 Related Work

**Concept learning.** In recent years, a significant strand of research has focused on employing labeled data to learn concepts in generative models' latent space for image editing and manipulation [1–6]. Concurrently, another independent research trajectory has been exploring unsupervised concept discovery and its potential to learn more compositional and transferable models [7–11]. Concurrently, a plethora of work has been dedicated to extracting interpretable concepts from high-dimensional data such as images. Concept-bottleneck [12] first predicts a set of human-annotated concepts as an intermediate stage and then predicts the task labels from these intermediate concepts. This paradigm has attracted a large amount of follow-up work [13, 29–33]. A recent surge of pre-trained multimodal models (e.g., CLIP [17]) can explain the image concepts through text directly [14–16].

**Latent variable identification.** Complex real-world data distributions often possess a hierarchical structure among their underlying latent variables. The identification conditions of latent hierarchical structures are investigated under the assumption that the latent variables are continuous and influence each other through linear functions [18–20] and nonlinear functions [34]. In addition, prior work [21, 26, 22, 23] studies fully discrete cases and thus falls short of modeling the continuous observed variables like images. To identify latent variables under nonlinear transformations, a line of work [35–38] assumes the availability of auxiliary information (e.g., domain/class labels) and that the latent variables' probability density functions have sufficiently different derivatives over domains/classes. Another line of studies [39, 40] refrains from the auxiliary information by assuming sparsity and mechanistic independence, disregarding causal structures among the latent variables.

Please refer to Section A1 for more extensive related work and discussion.

## 3 Discrete Hierarchical Models

**Data-generating process.** We formulate the data-generating process as the following latent-variable model. Let $\mathbf{x}$ denote the continuous observed variables $\mathbf{x} := [x_1, \cdots, x_n] \in \mathcal{X} \subset \mathbb{R}^{d_\mathbf{x}}$ which represents the high-dimension data we work with in practice (e.g., images). [1] Let $\mathbf{d} := [d_1, \cdots, d_{n_d}]$ be discrete latent variables that are direct parents to $\mathbf{x}$ (as shown in Figure 1(b)) and take on values from finite sets, i.e., $d_i \in \Omega_i^{(d)}$ for all $i \in [d_i]$ and $2 \leq \left|\Omega_i^{(d)}\right| < \infty$. We denote the joint domain as $\Omega^{(d)} := \Omega_1^{(d)} \times \cdots \times \Omega_{n_d}^{(d)}$. These discrete variables are potentially related to each other causally (e.g., $d_4$ and $d_5$ in Figure 1(c)) or via higher-level latent variables (e.g., $d_1$ and $d_2$ in Figure 1(c)). Let $\mathbf{c} := [c_1, \cdots, c_{n_c}] \in \mathcal{C} \subset \mathbb{R}^{d_c}$ be continuous latent variables that represent the continuous information conveyed in observed variables $\mathbf{x}$. The generating process is defined in Equation 1 and illustrated in Figure 1(a).

$$\mathbf{x} := g(\mathbf{d}, \mathbf{c}), \tag{1}$$

where we denote the generating function with $g : [\mathbf{d}, \mathbf{c}] \mapsto \mathbf{x}$. We denote the resultant bipartite graph from $[\mathbf{d}, \mathbf{c}]$ to $\mathbf{x}$ as $\Gamma$. In this context of image generation, the discrete subspace $\mathbf{d}$ gives a description of concepts present in the image $\mathbf{x}$ (e.g., a dog's appearance, background objects), and the continuous subspace $\mathbf{c}$ controls extents/degrees of specific attributes (e.g., sizes, lighting, and angles).

**Discrete hierarchical models.** As discussed above, discrete variables $d_1, \ldots, d_{n_d}$ represent distinct concepts that may be dependent either causally or purely statistically via higher-level concepts, as visualized in Figure 1(c). For instance, the dog's eye features and nose features are dependent, which a higher-level concept "breeds" could explain. We denote such higher-level latent discrete variables as $\mathbf{z} := [z_1, \cdots, z_{n_z}]$, where $z_i \in \Omega_i^{(z)}$ for all $i \in [z_i]$ and $2 \leq \left|\Omega_i^{(z)}\right| < \infty$ and $\Omega^{(z)} := \Omega_1^{(z)} \times \cdots \times \Omega_{n_z}^{(z)}$. Graphically, these variables $\mathbf{z}$ are not directly adjacent to observed

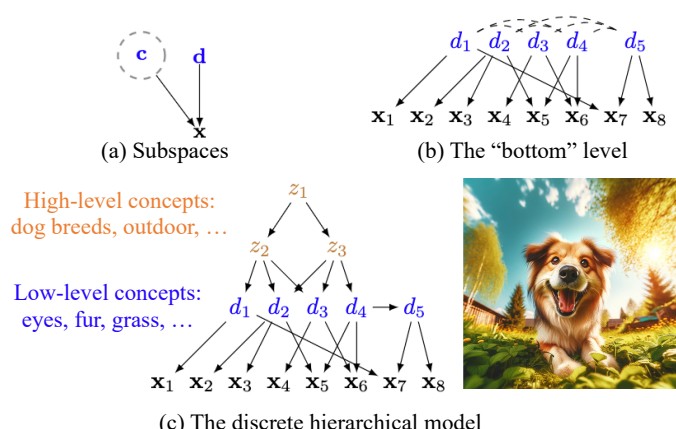

(a) Subspaces

(b) The "bottom" level

High-level concepts: dog breeds, outdoor, …

Low-level concepts: eyes, fur, grass, …

(c) The discrete hierarchical model

Figure 1: **Latent hierarchical graphs.** The dashed circle in (a) indicates that the continuous variable $\mathbf{c}$ can be viewed as an exogenous variable. Dashed edges in (b) indicate potential statistical dependence.

variables $\mathbf{x}$ (Figure 1(c)). High-level discrete variables $\mathbf{z}$ may constitute a hierarchical structure until the dependence in the system is fully explained. Since the discrete variables encode major semantic concepts in the data, this work primarily concerns discrete variables $\mathbf{d}$ and its underlying causal structure. The continuous subspace $\mathbf{c}$ can be viewed as exogenous variables and is often omitted in the causal graph (e.g., Figure 1(b)). We leave identifying continuous attributes in $\mathbf{c}$ as future work.

---

[1] We use the unbolded symbol $x_i$ to distinguish each observed variable $x_i$ from the collection $\mathbf{x}$. Our theory allows $x_i$ to be multi-dimensional.

Given this, we define the discrete hierarchical model as follows. The discrete hierarchical model (Figure 1(c)) $\mathcal{G} := (\mathbf{E}, \mathbf{V})$ is a DAG that comprises discrete latent variables $d_1, \cdots, d_{n_d}, z_1, \cdots, z_{n_z}$. We denote that directed edge set with $\mathbf{E}$ and the collection of all variables with $\mathbf{V} := \{\mathbf{D}, \mathbf{Z}\}$, where $\mathbf{D}$ and $\mathbf{Z}$ are vectors $\mathbf{d}$ and $\mathbf{z}$ in a set form and all leaf variables in $\mathcal{G}$ belong to $\mathbf{D}$. We assume the distribution over all variables $\mathbf{V}$ respects the Markov property with respect to the graph $\mathcal{G}$. We denote all parents and children of a variable with $\text{Ch}(\cdot)$ and $\text{Pa}(\cdot)$ respectively and define the neighbors as $\text{ne}(\cdot) := \text{Ch}(\cdot) \cup \text{Pa}(\cdot)$. We say a variable set $\mathbf{A}$ are pure children of $\mathbf{B}$, iff $\text{Pa}_{\mathcal{G}}(\mathbf{A}) = \cup_{A_i \in \mathbf{A}} \text{Pa}_{\mathcal{G}}(A_i) = \mathbf{B}$ and $\mathbf{A} \cap \mathbf{B} = \emptyset$. As shown in Figure 1(c), $x_1$ is a pure child of $d_1$.

**Objectives.** Formally, given only the observed distribution $p(\mathbf{x})$, we aim to:

1. identify discrete variables $\mathbf{d}$ and the bipartite graph $\Gamma$;
2. identify the hierarchical causal structure $\mathcal{G}$.

# 4 Identification of Discrete Latent Hierarchical Models

We present our theoretical results on the identifiability of discrete latent variables $\mathbf{d}$ and the bipartite graph $\Gamma$ in Section 4.2 (i.e., Objective 1) and the hierarchical model $\mathcal{G}$ in Section 4.3 (i.e., Objective 2).

**Additional notations.** We denote the set containing components of $\mathbf{x}$ with $\mathbf{X}$, the set of all variables with $\mathbf{V}^* := \mathbf{V} \cup \mathbf{X}$, the entire edge set with $\mathbf{E}^* := \mathbf{E} \cup \Gamma$, and the entire causal model with $\mathcal{G}^* := (\mathbf{V}^*, \mathbf{E}^*)$. As the true generating process involves $\mathbf{d}, \mathbf{c}, g, \Gamma$, and $\mathcal{G}$ (defined in Section 3), we define their statistical estimates with $\hat{\mathbf{d}}, \hat{\mathbf{c}}, \hat{g}$, and $\hat{\Gamma}$ through maximum likelihood estimation over the full population $p(\mathbf{x})$ while respecting conditions on the true generating process. We use $|\text{Supp}(\mathbf{L})|$ for the cardinality of a discrete variable set $\mathbf{L}$'s support (all joint states) and $\mathbf{P}_{\mathbf{A},\mathbf{B}}$ for the joint probability table whose two dimensions are the states of discrete variable sets $\mathbf{A}$ and $\mathbf{B}$ respectively.

## 4.1 General Conditions for Discrete Latent Models

It is well known that causal structures cannot be identified without proper assumptions. For instance, one may merge two adjacent discrete variables $d_1 \in \Omega_1^{(d)}$ and $d_2 \in \Omega_2^{(d)}$ into a single variable $\tilde{d} \in \Omega_1^{(d)} \times \Omega_2^{(d)}$ while preserving the observed distribution $p(\mathbf{x})$. We introduce the following basic conditions on the discrete latent model to eliminate such ill-posed situations.

**Condition 4.1** (General Latent Model Conditions)**.**

  i *[Non-degeneracy]:* $\mathbb{P}(\mathbf{d} = k_1, \mathbf{z} = k_2) > 0$, *for all* $(k_1, k_2) \in \Omega^{(d)} \times \Omega^{(z)}$; *for all variable* $v \in \mathbf{V}^*$, $\mathbb{P}(v|Pa(v) = k_1) \neq \mathbb{P}(v|Pa(v) = k_2)$ *if* $k_1 \neq k_2$.

  ii *[No-twins]: Distinct latent variables have distinct neighbors* $ne(v_1) \neq ne(v_2)$, *if* $v_1 \neq v_2 \in \mathbf{V}$.

  iii *[Maximality]: There is no DAG* $\tilde{\mathcal{G}}^* := (\tilde{\mathbf{V}}^*, \tilde{\mathbf{E}}^*)$ *resulting from splitting a latent variable in* $\mathcal{G}^*$ *(i.e., turning* $z_i$ *into* $\tilde{z}_{i,1}$ *and* $\tilde{z}_{i,2}$ *with identical neighbors and cardinality* $|\Omega_i^z| = |\tilde{\Omega}_{i,1}^z| + |\tilde{\Omega}_{i,2}^z|$ *), such that* $\mathbb{P}\left(\tilde{\mathbf{V}}^*\right)$ *is Markov w.r.t.* $\tilde{\mathcal{G}}^*$ *and* $\tilde{\mathcal{G}}^*$ *satisfies ii.*

**Discussion.** Condition 4.1 is a necessary set of conditions for identifying latent discrete models, which is employed and discussed extensively [24, 41]. Intuitively, Condition 4.1-i excludes dummy discrete states and graph edges that exert no influence on the observed variables $\mathbf{x}$. Condition 4.1-ii,iii constrain the latent model to be the most informative graph without introducing redundant latent variables, thus forbidding arbitrary merging and splitting over latent variables.

## 4.2 Discrete Component Identification

We show with access to only the observed data $\mathbf{x}$, we can identify each discrete component $d_i$ up to permutation indeterminacy (Definition 4.2) and a corresponding bipartite graph equivalent to $\Gamma$.

**Definition 4.2** (Component-wise Identifiability)**.** Variables $\mathbf{d} \in \mathbb{N}^{n_d}$ and $\hat{\mathbf{d}} \in \mathbb{N}^{n_d}$ are identified component-wise if there exists a permutation $\pi$, such that $\hat{d}_i = h_i(d_{\pi(i)})$ with invertible function $h_i$.

That is, our estimation $\hat{d}_i$ captures full information of $d_{\pi(i)}$ and no information from $d_j$ such that $j \neq \pi(i)$. [2] The permutation is a fundamental indeterminacy for disentanglement [37, 38, 36, 24].

---

[2] We use "components" to refer to individual discrete variables $d_i$ in the vector $\mathbf{d}$.

**Remarks on the problem.** A large body of prior work [37, 35, 42] requires continuous or even differentiable density function over all latent variables and domain/class labels or counterfactual counterparts to generate variation. Thus, their techniques do not transfer naturally to our latent space with both continuous and discrete parts $[\mathbf{c}, \mathbf{d}]$ and no supervision of any form. With a similar goal, Kivva et al. [24] assumes access to an oracle (Definition A2.1) to the mixture distribution over $p(\mathbf{x})$, which is not directly available in the general case here. Kivva et al. [41] assumes a specific parametric generating process, whereas we focus on a generic non-parametric generative model (Equation 1).

**High-level description of our proposed approach.** We decompose the problem into two tractable subproblems: 1) extracting the global discrete state $\mathbf{d}$ from the mixing with the continuous variable $\mathbf{c}$; 2) further identifying each discrete component $d_i$ from the mixing with other discrete components $d_j$ ($i \neq j$) and the causal graph $\Gamma$. For 1), we show that, perhaps surprisingly, minimal conditions on the generating function $g$ suffice to remove the information of $\mathbf{c}$ and thus identify the global state of $\mathbf{d}$. For 2), we observe that the identification results in 1) can be viewed as a mixture oracle over $p(\mathbf{x})$, which enables us to employ techniques from Kivva et al. [24] to solve the problem.

We introduce key conditions and formal theoretical statements as follows.

**Condition 4.3** (Discrete Components Identification)**.**

- *i [Connected Spaces] The continuous support $\mathcal{C} \subset \mathbb{R}^{n_c}$ is closed and connected.*

- *ii [Invertibility & Continuity]: The generating function $g$ in equation 1 is invertible, and for any fixed $\mathbf{d}$, $g(\mathbf{d}, \cdot)$ and its inverse are continuous.*

- *iii [Non-Subset Observed Children]: For any pair $d_i$ and $d_j$, one's observed children are not the subset of the other's, $Ch_\Gamma(d_i) \not\subset Ch_\Gamma(d_j)$.*

**Discussion on the conditions.** Condition 4.3-i requires the continuous support $\mathcal{C}$ to be regular in contrast with the discrete variable's support. Intuitively, the continuous variable $\mathbf{c}$ often controls the extents/degrees of specific attributes (e.g., sizes, lighting, and angles) and takes values from connected spaces. For instance, "lightning" ranges from the lowest to the highest intensity continuously. Condition 4.3-ii ensures the generating process preserves latent variables' information [37, 35, 42, 43]. Thanks to the high dimensionality, images often have adequate capacity to meet this condition. For instance, the image of a dog contains a detailed description of the dog's breed, shape, color, lighting intensity, and angles, all of which are decodable from the image. Condition 4.3-iii ensures that each latent component should exhibit sufficiently distinguishable influences on the observed variable $\mathbf{x}$. Practically, this condition indicates that the lowest-level concepts influence diverse parts of the image. These concepts are often atomic, such as a dog's ear, eyes, or even finer, which often don't overlap. This condition is adopted in prior work [24, 41] and related to the notation of sparsity. Along this line, prior work [44–46] assumes pure observed children for each discrete variable, which is strictly stronger. Recent work [47] assumes each latent variable is connected to a unique set of observed variables. This condition implies Condition 4.3-iii because if $z_0$'s children form a subset of $z_1$'s children, then one cannot find a subset of observed variables whose parent is $z_0$ alone.

**Theorem 4.4** (Discrete Component Identification)**.** *Under the generating process in Equation 1 and Condition 4.3-ii, the estimated discrete variable $\hat{\mathbf{d}}$ and the true discrete variable $\mathbf{d}$ are equivalent up to an invertible function, i.e., $\hat{\mathbf{d}} = h(\mathbf{d})$ with $h(\cdot)$ invertible. Moreover, if Condition 4.1 and Condition 4.3-iii further hold, we attain component-wise identifiability (Definition 4.2) and the bipartite graph $\Gamma$ up to permutation of component indices.*

**Proof sketch.** Intuitively, each state of the discrete subspace $\mathbf{d}$ indexes a manifold $g(\mathbf{d}, \cdot) : \mathbf{c} \mapsto \mathbf{x}$ that maps the continuous subspace $\mathbf{c}$ to the observed variable $\mathbf{x}$. These manifolds do not intersect in the observed variable space $\mathcal{X}$ regardless of however close they may be to each other, thanks to the invertibility of the generating function $g$ (Condition 4.3-ii). This leaves a sufficient footprint in $\mathbf{x}$ for us to uniquely identify the manifold it resides in, giving rise to the identifiability of $\mathbf{d}$. This reveals the discrete state of each realization of $\mathbf{x}$ and equivalently the joint distribution $p(\tilde{d}, \mathbf{x})$ where we merge all components in $\mathbf{d}$ into a discrete variable $\tilde{d}$. Identifying this joint distribution enables the application of tensor decomposition techniques [24] to disentangle the global state $\hat{d}$ into individual discrete components $d_i$ and the causal graph $\Gamma$, under Condition 4.1 and Condition 4.3-iii.

### 4.3 Hierarchical Model Identification

We show that we can identify the underlying hierarchical causal structure $\mathcal{G}$ that explains the dependence among low-level discrete components $d_i$ that we identify in Theorem 4.4.

**Remarks on the problem.** Benefiting from the identified discrete components in Theorem 4.4, we employ $\mathbf{d}$ as observed variables to identify the discrete latent hierarchical model $\mathcal{G}$. Although discrete latent hierarchical models have been under investigation for an extensive period, existing results mostly assume relatively strong graphical conditions – the causal structures are either trees [26, 21, 22] or multi-level DAGs [23, 48], which can be restrictive in capturing the complex interactions among latent variables among different hierarchical levels. Separately, recent work [19, 20] has exhibited more flexible graphical conditions for linear, continuous latent hierarchical models. For instance, prior work [20] allows for multiple directed paths of disparate edge numbers within a variable pair and potential non-leaf observed variables. Unfortunately, their techniques hinge on linearity and cannot directly apply to discrete models of high nonlinearity.

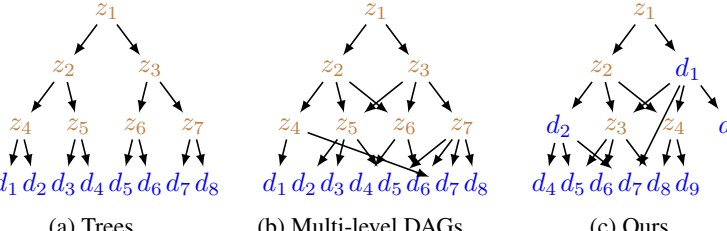

(a) Trees.      (b) Multi-level DAGs.      (c) Ours.

Figure 2: **Graphical comparison.** Tree Structures permit one undirected path between any two variables. Multi-level DAGs require partitioning variables into levels with edges only between adjacent levels. Our conditions allow multiple paths between variables across levels and include non-leaf observed variables.

**High-level description of our approach.** The central machinery in prior work [19, 20] is Theorem A2 [49], which builds a connection between easily computable statistical quantities (i.e., sub-covariance matrix ranks) and local latent graph information. Dong et al. [20] utilize a graph search algorithm to piece together these local latent graph structures to identify the entire hierarchical model. Ideally, if we can access these local latent structures in the discrete model, we can apply the same graph search procedure and theorems to identify the discrete model. Nevertheless, Theorem A2 relies on linearity (i.e., each causal edge represents a linear function), which doesn't hold in the discrete case. We show that interestingly, Theorem A2 can find a counterpart in the discrete case (Theorem 4.8), despite the absence of linearity. Since given the graphical information from Theorem A2, the theory in Dong et al. [20] is independent of statistical properties, we can utilize flexible conditions and algorithm therein by obtaining the same graphical information with Theorem 4.8.

To present Theorem 4.8, we introduce non-negative rank $\text{rank}_+(\cdot)$ [50] (Definition 4.5), and t-separation [51] (Definition 4.7) as follows.

**Definition 4.5** (Non-negative Rank). The non-negative rank of non-negative $A \in \mathbb{R}_+^{m \times n}$ is equal to the smallest $p$ for which there exist $\mathbf{B} \in \mathbb{R}_+^{m \times p}$ and $\mathbf{C} \in \mathbb{R}_+^{p \times n}$ such that $\mathbf{A} = \mathbf{BC}$.

**Definition 4.6** (Treks). A trek $T_{i,j}$ in a DAG from vertex $i$ to $j$ consists of a directed path $P_{ki}$ from $k$ to $i$ and a direct path $P_{kj}$ from $k$ to $j$, where we refer to $P_{ki}$ as the $i$ side and $P_{kj}$ as the $j$ side.

**Definition 4.7** (T-separation). Let $\mathbf{A}$, $\mathbf{B}$, $\mathbf{C_A}$, and $\mathbf{C_B}$ be subsets (not necessarily disjoint) of vertices in a DAG. Then $(\mathbf{C_A}, \mathbf{C_B})$ t-separates $\mathbf{A}$ and $\mathbf{B}$ if every trek from $\mathbf{A}$ to $\mathbf{B}$ passes through either a vertex in $\mathbf{C_A}$ on the $\mathbf{A}$ side of the trek or a vertex $\mathbf{C_B}$ on the $\mathbf{B}$ side of the trek.

Intuitively, a trek is a path containing at most one fork structure and no collider structures. It is known that one can formulate d-separation as a special form of t-separation (see Theorem A1). Thus, t-separation is at least as informative as d-separation. As detailed in Dong et al. [20], t-separation can provide more information when latent variables are involved, benefiting from Theorem A2 [49].

**Theorem 4.8** (Implication of Rank Information on Latent Discrete Graphs). *Given two sets of variables $\mathbf{A}$ and $\mathbf{B}$ from a non-degenerate, faithful (Condition 4.1-i, Condition 4.10-i) discrete model $\mathcal{G}$, it follows that $\text{rank}_+(\mathbf{P_{A,B}}) = \min\{|Supp(\mathbf{L})| : \text{a partition } (\mathbf{L}_1, \mathbf{L}_2) \text{ t-separates } \mathbf{A} \text{ and } \mathbf{B} \text{ in } \mathcal{G}\}.$*

**Example.** Suppose every variable in Figure 2(a) is binary, then for $\mathbf{A} = \{d_1, d_2, d_3\}$, $\mathbf{B} = \{d_3, \ldots, d_8\}$, $\text{rank}_+(\mathbf{P_{A,B}}) = 4$ since $\mathbf{A}$ and $\mathbf{B}$ are t-separated by $\{d_3, z_4\}$ with 4 states.

**Discussion.** Parallel to Theorem A2 [49] for linear models, Theorem 4.8 acts as an oracle to reveal the minimal t-separation set's cardinality between any two variable sets in *discrete* models beyond linearity. This enables us to infer the latent graph structure from only the observed variables' statistical information. To the best of our knowledge, Theorem 4.8 is the first to establish this connection and can be of independent interest for learning latent discrete models in future work. Although the computation of non-negative ranks can be expensive [50], existing work [52, 53] demonstrates that regular rank tests are decent substitutes, we observe in our synthetic data experiments (Section 5).

We present the identification conditions for discrete models as follows (Condition 4.10).

**Definition 4.9** (Atomic Covers). Let $\mathbf{A} \subset \mathbf{V}$ be a set of variables in $\mathcal{G}$ with $|\text{Supp}(\mathbf{A})| = k$, where $t$ of the $k$ states belong to observed variables $d_i$, and the remaining $k - t$ are from latent variables $z_j$. $\mathbf{A}$ is an atomic cover if $\mathbf{A}$ contains a single observed variable, or if the following conditions hold:

(i) There exists a set of atomic covers $\mathcal{C}$, with $|\text{Supp}(\mathcal{C})| \geq k+1-t$, such that $\cup_{\mathbf{C} \in \mathcal{C}} \mathbf{C} \subseteq \text{PCh}_{\mathcal{G}}(\mathbf{A})$ and $\forall \mathbf{C_1}, \mathbf{C_2} \in \mathcal{C}, \mathbf{C_1} \cap \mathbf{C_2} = \emptyset$.

(ii) There exists a set of covers $\mathcal{N}$, with $|\text{Supp}(\mathcal{N})| \geq k+1-t$, such that every element in $\cup_{\mathbf{N} \in \mathcal{N}} \mathbf{N}$ is a neighbour of $\mathbf{V}$ and $(\cup_{\mathbf{N} \in \mathcal{N}} \mathbf{N}) \cap (\cup_{\mathbf{C} \in \mathcal{C}} \mathbf{C}) = \emptyset$.

(iii) There does not exist a partition of $\mathbf{A} = \mathbf{A_1} \cup \mathbf{A_2}$ such that both $\mathbf{A_1}$ and $\mathbf{A_2}$ are atomic covers.

**Example.** In Figure 2 (c), $\{z_2\}$ is an atomic cover if its pure child $\{d_2\}$ and its neighbors $\{z_1, z_3, z_4\}$ possess more than $\text{Supp}(z_2) + 1$ states separately. Otherwise, $\{z_2, d_1\}$ can be an atomic cover if (some of) pure children $\{z_3, z_4\}$ and neighbors $\{z_1, d_2, d_3\}$ possess $\text{Supp}(z_2) + 1$ states separately.

**Condition 4.10** (Discrete Hierarchical Model Conditions).

   *i [Faithfulness] All the conditional independence relations are entailed by the DAG.*

   *ii [Basic Graphical Conditions] Each latent variable $z \in \mathbf{Z}$ corresponds to a unique atomic cover in $\mathcal{G}$ and no $z$ is involved in any triangle structure (i.e., three mutually adjacent variables).*

   *iii [Graphical Condition on Colliders] In a latent graph $\mathcal{G}$, if (i) there exists a set of variables $\mathbf{C}$ such that every variable in $\mathbf{C}$ is a collider of two atomic covers $\mathbf{L_1}$, $\mathbf{L_2}$, and denote by $\mathbf{A}$ the minimal set of variables that d-separates $\mathbf{L_1}$ from $\mathbf{L_2}$, (ii) there is a latent variable in $\mathbf{L_1}, \mathbf{L_2}, \mathbf{C}$ or $\mathbf{A}$, then we must have $|\text{Supp}(\mathbf{C})| + |\text{Supp}(\mathbf{A})| \geq |\text{Supp}(\mathbf{L_1})| + |\text{Supp}(\mathbf{L_2})|$.*

**Discussion on the conditions.** Condition 4.10-i is known as the faithfulness condition widely adopted for causal discovery [51, 24, 54, 18], which attributes statistical independence to graph structures rather than unlikely coincidence [55, 51]. In linear models, Dong et al. [20] introduce atomic covers (Definition 4.9) to represent a group of indistinguishable variables. In the discrete case, an atomic cover consists of indistinguishable latent states, which we merge into a single latent discrete variable (Condition 4.1-ii). Intuitively, we treat each state as a separate variable and merge those belonging to the same atomic cover at the end of the identification procedure. This handles discrete variables of arbitrary state numbers, in contrast with the binary or identical support assumptions [22, 23], which we use as an alternative condition in Theorem A12. Condition 4.10-ii requires each atomic cover to possess sufficiently many children and neighbors to preserve its influence while avoiding problematic triangle structures to ensure the uniqueness of its influence. In contrast, existing work [23] assumes at least three pure children for each latent variable, amounting to six times more states. Condition 4.10-iii ensures adequate side information (large $|\mathbf{A}|$) to discover latent colliders $\mathbf{C}$, admitting graphs more general than tree structures [21, 26, 22] (i.e., no colliders). Overall, our model encompasses a rich class of latent structures more complex than tree structures and multi-level DAGs [23] (Figure 2).

Following Dong et al. [20], we introduce the minimal-graph operator $\mathcal{O}_{\min}$ (Definition 4.11 and Figure A1), which merges certain redundancy structures that rank information cannot distinguish.

**Definition 4.11** (Minimal-graph Operator [19, 20]). We can merge atomic covers $\mathbf{L}$ into $\mathbf{P}$ in $\mathcal{G}$ if (i) $\mathbf{L}$ is a pure child of $\mathbf{P}$, (ii) all elements of $L$ and $P$ are latent and $|\text{Supp}(\mathbf{L})| = |\text{Supp}(\mathbf{P})|$, and (iii) the pure children of $\mathbf{L}$ form a single atomic cover, or the siblings of $\mathbf{L}$ form a single atomic cover. We denote such an operator as the minimal-graph operator $\mathcal{O}_{\min}(\mathcal{G})$.

**Theorem 4.12** (Discrete Hierarchical Identification). *Suppose the causal model $\mathcal{G}$ satisfies Condition 4.1 and Condition 4.10 We can identify $\mathcal{G}$ up to the Markov equivalence class of $\mathcal{O}_{\min}(\mathcal{G})$.*

**Proof sketch.** As discussed above, Theorem 4.8 gives a graph structure oracle equivalent to Theorem A2, which we leverage to prove Theorem 4.12. Besides the rank test, the major distinction

Table 1: **F1 scores for our method and the baseline Kivva et al. [24]** . Figure A2 exhibits the graphs.

|  | Graph 1 | Graph 2 | Graph 3 | Graph 4 | Graph 5 | Graph 6 | Graph 7 | Graph 8 | Graph 9 |
|---|---|---|---|---|---|---|---|---|---|
| Baseline | $0.67 \pm 0.0$ | $0.69 \pm 0.1$ | $0.67 \pm 0.0$ | $0.67 \pm 0.2$ | $0.63 \pm 0.0$ | $0.65 \pm 0.0$ | $0.67 \pm 0.0$ | $0.65 \pm 0.0$ | $0.63 \pm 0.0$ |
| Ours | $0.94 \pm 0.1$ | $0.98 \pm 0.1$ | $0.94 \pm 0.0$ | $0.98 \pm 0.2$ | $0.94 \pm 0.1$ | $0.93 \pm 0.0$ | $0.93 \pm 0.1$ | $0.96 \pm 0.0$ | $0.93 \pm 0.1$ |

Table 2: **F1 scores for our method and the baseline Dong et al. [20]**. Figure A3 exhibits the graphs.

|  | Graph 1 | Graph 2 | Graph 3 | Graph 4 | Graph 5 | Graph 6 | Graph 7 |
|---|---|---|---|---|---|---|---|
| Baseline | $0.24 \pm 0.3$ | $0.48 \pm 0.0$ | $0.33 \pm 0.2$ | $0.63 \pm 0.1$ | $0.0 \pm 0.0$ | $0.55 \pm 0.1$ | $0.0 \pm 0.0$ |
| Ours | $1.0 \pm 0.0$ | $1.0 \pm 0.0$ | $0.73 \pm 0.0$ | $0.73 \pm 0.0$ | $0.75 \pm 0.0$ | $0.95 \pm 0.0$ | $1.0 \pm 0.0$ |

between Theorem A2 and Theorem 4.8 is that the former returns the number of variables in the minimal t-separation set whereas the latter returns the number of states. Applying the search algorithm from Dong et al. [20] alongside our rank test from Theorem 4.8 to a discrete model $\mathcal{G}$ results in a graph $\tilde{\mathcal{G}}$. In $\tilde{\mathcal{G}}$, each latent variable $z$ in $\mathcal{G}$ is split into a set of variables $\tilde{z}^{(1)}, \ldots, \tilde{z}^{(|\mathrm{Supp}(z)|)}$ as an atomic cover, with the set size equal to the state number of $z$. We can then reconstruct the original graph $\mathcal{G}$ from $\tilde{\mathcal{G}}$ by merging these atomic covers into discrete variables. We present our algorithm in Algorithm 1 and highlight the differences from that in Dong et al. [20].

Our techniques can also utilize the identical support condition (e.g., binary latent variables) [23, 22] for identification under slightly different conditions. We present the results in Theorem A12.

## 5  Synthetic Data Experiments

**Experimental setup.** We generate the hierarchical model $\mathcal{G}$ with randomly sampled parameters, and follow [24] to build the generating process from $\mathbf{d}$ to the observed variables $\mathbf{x}$ (i.e., graph $\Gamma$) by a Gaussian mixture model. The graphs are exhibited in Figure A2 and Figure A3 in Appendix A4. We follow Dong et al. [20] to use F1 score for evaluation. More details can be found in Appendix A4.

**Results and discussion.** We choose Kivva et al. [24] as our baseline because it is the only method we know designed to learn a non-parametric, discrete latent model from continuous observations. We evaluate both methods on graphs in Figure A2. As shown in Table 1 and Table 2, our method consistently achieves near-perfect scores, while the baseline, despite correctly identifying $\Gamma$ and directing edges among $\mathbf{d}$ components, cannot handle higher-level latent variables.

To verify Theorem 4.8, we evaluate Algorithm 1 and a baseline [20] on graphs satisfying the conditions on $\mathcal{G}$ (i.e., purely discrete models in Figure A3). Our method performs well on graphs that meet conditions of Theorem A12 and achieves decent scores on graphs that do not (Figure A3 (c) and (e)). The significant margins over the baseline validate Theorem 4.8 and Theorem A12.

## 6  Interpretations of Latent Diffusion

In this section, we present a novel interpretation of latent diffusion (LD) [28] from the perspective of our hierarchical concept learning framework. Concretely, the diffusion training objective can be viewed as performing denoising autoencoding at different noise levels [56, 57]. Denoising autoencoders [58, 59] and variants [60, 61] have shown the capability of extracting high-level, semantic representations as their encoder output. In the following, we adopt this perspective to interpret the diffusion model's representation (i.e., the UNet encoder output) through our hierarchical model, which connects the noise level and the hierarchical level of the latent representation in our causal model. For brevity, we refer to the diffusion model encoder's output as diffusion representation.

**Discrete variables and representation embeddings.** In practice, discrete variables are often modeled as embedding vectors from a finite dictionary (e.g., wording embeddings). Therefore, although diffusion representation is not discrete, we can interpret it as an ensemble of embeddings of involved discrete variables. Park et al. [62] empirically demonstrates that one can indeed decompose the diffusion representation into a finite set of basis vectors that carry distinct semantic information, which can be viewed as the concept embedding vectors.

**Vector-quantization.** Given an image $\mathbf{x}$, LD first discretizes it with a vector-quantization generative adversarial network (VQ-GAN) [63]:$\mathbf{d} = f_{\mathrm{VQ}}(\mathbf{x})$. Through the lens of our framework, VQ-GAN represents the image with a rich but finite set of embeddings of bottom-level concepts $\mathbf{d}$ and discards nuances in the continuous representation $\mathbf{c}$, inverting the generation process in Equation 1.

**Denoising objectives.** As discussed, diffusion training can be viewed as denoising the corrupted embedding $\tilde{\mathbf{d}}$ to restore noiseless $\mathbf{d}$ [57–59, 64] for a designated denoising model $f_t$ at noise level $t$:

$$\arg\max_{f_t} \mathbb{E}_{\mathbf{d}, \tilde{\mathbf{d}}_t \sim \mathbb{Q}_t(\tilde{\mathbf{d}}_t | \mathbf{d})} \left[ \mathbb{P}_{f_t}(\mathbf{d} | \tilde{\mathbf{d}}_t, \mathbf{y}) \right], \quad (2)$$

where $\mathbf{y}$ denotes the text prompt. Under this objective, the model is supposed to compress the noisy view $\tilde{\mathbf{d}}_t$ to extract a clean, high-level representation, together with additional information from the text $\mathbf{y}$, to reconstruct the original embedding $\mathbf{d}$. Formally, the denoising model $f_t := f_{\text{dec},t} \circ f_{\text{enc},t}$ performs auto-encoding

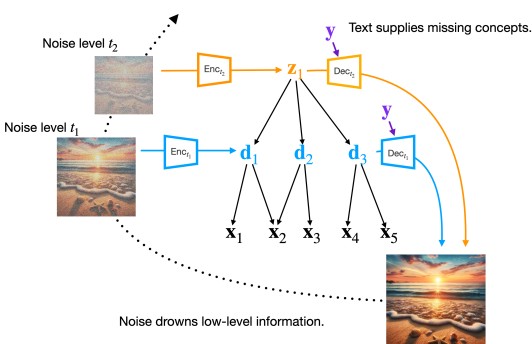

Figure 3: **Diffusion models estimate the latent hierarchical model.** Different noise levels correspond to different concept levels. To avoid cluttering, we leave out vector quantization.

$\mathbf{z}_{\mathcal{S}(t)} = f_{\text{enc},t}(\tilde{\mathbf{d}}_t)$ and $\hat{\mathbf{d}} = f_{\text{dec},t}(\mathbf{z}_{\mathcal{S}(t)}, \mathbf{y})$, where we use $\mathcal{S}(t)$ to indicate the dependence on the noise level $t$. We can view the compressed representation $\mathbf{z}_{\mathcal{S}(t)}$ as a set of high-level latent variables in the hierarchical model: *the encoder $f_{enc,t}$ maps the noisy view $\tilde{\mathbf{d}}_t$ to high-level latent variables $\mathbf{z}_{\mathcal{S}_t}$ and the decoder $f_{dec,t}$ assimilates the text information $\mathbf{y}$ and reconstructs the original view $\mathbf{d}$.* In practice, $f_t$ is implemented as a single model (e.g., UNets) paired with time embeddings. We visualize this process in Figure 3.

**Noise levels and hierarchical levels.** Intuitively, the noise level controls the amount of semantic information remaining in $\tilde{\mathbf{d}}_t$. For instance, a high noise level $t$ drowns the bulk of the low-level concepts in $\mathbf{d}$, leaving only sparse high-level concepts in $\tilde{\mathbf{d}}_t$. In this case, the diffusion representation $\mathbf{z}_{\mathcal{S}(t)}$ estimates a high concept level in the hierarchical model. In Figure 3, a high noise level may destroy low-level concepts, such as the sand texture and the waveforms, while preserving high-level concepts, such as the beach and the sunrise. In Section 7.1, we follow Park et al. [62] to demonstrate diffusion representation's semantic levels under different noise levels.

**Theory and practice.** We connect LD training and estimating latent variables in the hierarchical model in an intuitive sense. Our theory focuses on the fundamental conditions of the data-generating process and does not directly translate to guarantees for LD. That said, our conditions naturally have implications on the algorithm design. For instance, a sparsity constraint on the decoding model may facilitate the identification condition that variables influence each other sparsely (e.g., pure children in Condition 4.10). In Section 7.3, we show such constraints are beneficial for concept extraction. We hope that our new perspective can provide more novel insights into advancing practical algorithms.

# 7 Real-world Experiments

## 7.1 Discovering Hierarchical Concept Structures from Diffusion Models

In Figure 4, we extract concepts and their relationships from LD through our hierarchical model interpretation. Our recovery involves two stages: determining the concept level and identifying causal links. We add a textual concept, like "dog", into the prompt and identify the latest diffusion step that would render this concept properly. If "dog" appears in the image only when added at step 0 and "eye" appears when added from step 5, it indicates that "dog" is a higher-level concept than "eyes". After determining the levels of concepts, we intervene on a high-level concept and observe changes in low-level ones. No significant changes indicate no direct causal relationship. We explore the relationships among the concepts "dog", "tree", "eyes", "ears", "branch", and "leaf". Figure 4 presents the final recovered graph and intermediate results. See Section A5.3 for more investigation.

## 7.2 Diffusion Representation as Concept Embeddings

We support our interpretations in Section 6 that diffusion representation can be viewed as concept embeddings, and it corresponds to high-level concepts for high noise levels. Following Park et al. [62], we modify the diffusion representation along certain directions found unsupervisedly. We can observe that this manipulation gives rise to semantic concept changes rather than entangled corruption Figure 5. Editing the latent representation at early steps corresponds to shifting global concepts.

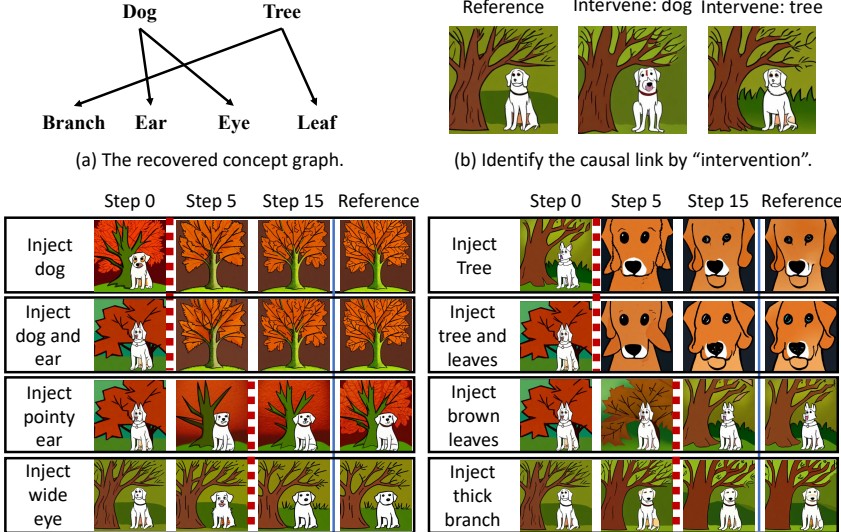

(a) The recovered concept graph.

(b) Identify the causal link by "intervention".

(c) Identify the concept level by the last effective diffusion step (red dotted lines).

Figure 4: **Recovering concepts and their relationships from LD.** (a) The final recovered concept graph among concepts "dog", "tree", "eyes", "ears", "branch", and "leaf". (b) Identifying causal links through "interventions". For example, we compare two prompts that vary in "dog": "a dog with wide eyes and a wilting tree with short branches, in a cartoon style" and "a big dog with wide eyes and a wilting tree with short branches, in a cartoon style". We observe significant changes in "eyes" but not in "branch", indicating a causal link between "dog" and "eyes" but not between "dog" and "branch". (c) Identifying concept levels by the last effective diffusion step. For example, we use the base prompt "a tree with long branches, in a cartoon style" and prepend "dog" at steps 0, 5, and 15. Only injecting "dog" at step 0 works. Similarly, injecting "wide eyes" works at both steps 0 and 5, indicating that "dog" is a higher-level concept than "eyes".

In Figure 5, the latent representation in earlier steps (step $T$) determines breeds (the top row), species (the middle row), and gender (the bottom row). In contrast, the latent representation in later steps (step $0.6T$) correlates with the dog collar, cat eyes, and shirt patterns. Implementation details and additional results are provided in Appendix A5.

### 7.3 Causal Sparsity for Concept Extraction

Recent work [65] shows that concepts can be extracted as low-rank parameter subspaces of LD models via LoRA [66]. This low-rankness limits the complexity of text-induced changes, resembling sparse influences from latent concepts to their descendants. Our theory suggests that different levels of concepts may require varying sparsity levels to capture. We present empirical evidence in Section A5.4. Motivated by this, we design an adaptive sparsity selection mechanism for capturing concepts at different levels. Inspired by Ding et al. [67], we implement a sparsity constraint on the LoRA dimensionality for the model to select the LoRA rank at each module automatically, benefiting concept extraction (see Appendix A5.4).

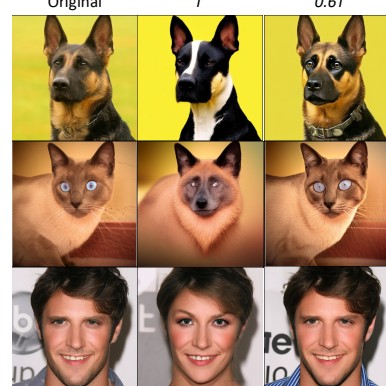

Figure 5: **Semantic latent space.** We modify the diffusion model's representation (UNet encoder's output) along principal directions at steps $T$ and $0.6T$. Structure changes indicate the semantics of the representation and manipulation at the early time $T$ induces global shifts. See more examples in Figure A9.

## 8 Conclusion

In this work, we cast the task of learning concepts as the identification problem of a discrete latent hierarchical model. Our theory provides conditions to guarantee the recoverability of discrete concepts. **Limitations:** Although our theoretical framework provides a lens for interpretation, our conditions do not directly guarantee diffusion's success, which would require nontrivial assumptions. Also, Algorithm 1 can be expensive for large graphs due to the dependency on the state count. We leave giving guarantees to diffusion models and efficient graph learning algorithms as future work.

**Acknowledgments.** We thank the anonymous reviewers for their valuable insights and recommendations, which have greatly improved our work. The work of L. Kong and Y. Chi is supported in part by NSF DMS-2134080. This material is based upon work supported by NSF Award No. 2229881, AI Institute for Societal Decision Making (AI-SDM), the National Institutes of Health (NIH) under Contract R01HL159805, and grants from Salesforce, Apple Inc., Quris AI, and Florin Court Capital. This research has been graciously funded by the National Science Foundation (NSF) CNS2414087, NSF BCS2040381, NSF IIS2123952, NSF IIS1955532, NSF IIS2123952; NSF IIS2311990; the National Institutes of Health (NIH) R01GM140467; the National Geospatial Intelligence Agency (NGA) HM04762010002; the Semiconductor Research Corporation (SRC) AIHW award 2024AH3210; the National Institute of General Medical Sciences (NIGMS) R01GM140467; and the Defense Advanced Research Projects Agency (DARPA) ECOLE HR00112390063. Any opinions, findings, and conclusions or recommendations expressed in this publication are those of the author(s) and do not necessarily reflect the views of the National Science Foundation, the National Institutes of Health, the National Geospatial Intelligence Agency, the Semiconductor Research Corporation, the National Institute of General Medical Sciences, and the Defense Advanced Research Projects Agency.

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

*Appendix for*

**"Learning Discrete Concepts in Latent Hierarchical Causal Models"**

Table of Contents

## A1   Related Work

**Concept learning.** In recent years, a significant strand of research has focused on employing labeled data to learn concepts in generative models' latent space for image editing and manipulation [1–6]. Concurrently, another independent research trajectory has been exploring unsupervised concept discovery and its potential to learn more compositional and transferable models, as shown in Burgess et al. [7], Locatello et al. [8], Du et al. [9, 10], Liu et al. [11]. These prior works focus on the empirical methodological development of concept learning by proposing novel neural network architectures and training objectives, with limited discussion on the theoretical aspect. In contrast, our work investigates the theoretical foundation of concept learning. Specifically, we formulate concept learning as an identification problem for a discrete latent hierarchical model and provide conditions under which extracting concepts is possible. Thus, the existing work and our work can be viewed as two complementary lines of research for concept learning. Concurrently, a plethora of work has been dedicated to extracting interpretable concepts from high-dimensional data such as images. Concept-bottleneck [12] first predicts a set of human-annotated concepts as an intermediate stage and then predicts the task labels from these intermediate concepts. This paradigm has attracted a large amount of follow-up work [13, 29–33]. A recent surge of pre-trained multimodal models (e.g., CLIP [17]) can explain the image concepts through text directly [14–16]. In contrast with these successes, our work focuses on the formulation of concept learning and theoretical guarantees.

**Latent hierarchical models.** Complex real-world data distributions often possess a hierarchical structure among their underlying latent variables. On the theoretical front, Xie et al. [18], Huang et al. [19], Dong et al. [20] investigate identification conditions of latent hierarchical structures under the assumption that the latent variables are continuous and influence each other through linear functions. Kong et al. [34] extends the functional class to the nonlinear case over continuous variables. Pearl [21], Zhang [26], Choi et al. [22], Gu and Dunson [23] study fully discrete cases and thus fall short of modeling the continuous observed variables like images. Specifically, Pearl [21], Zhang [26], Choi et al. [22] focus on the latent trees in which every pair of variables is connected through exactly one undirected path. Gu and Dunson [23] assume a multi-level DAG [68] in which variables can be partitioned into disjoint groups (i.e., levels), such that all edges are between adjacent levels, with the observed variables as the bottom level (i.e., leaf nodes). In contrast, we show that we can not only extract discrete components from continuous observed variables but also uncover higher-level concepts and their interactions. Our graphical conditions admit multiple paths within each pair of latent variables, flexible hierarchical structures that are not necessarily multi-level, and flat structures in which all latent variables are adjacent to observed variables [24]. On the empirical side, prior work [69] improves the inference model of vanilla VAEs by combining bottom-up data-dependent likelihood terms with prior generative distribution parameters. Zhao et al. [70] assign more expressive (deeper) neural modules to higher-level variables to learn a more disentangled generative model. Li et al. [71] present a VAE/clustering approach to empirically estimating latent tree structures. Leeb et al. [72] propose to feed latent variable partitions into different decoder neural network layers and remove the prior regularization term to enable high-quality generation. Like our work, Ross and Doshi-Velez [73] consider discrete latent variables. However, their focus is on empirical evaluation benchmarks and metrics, without touching on the theoretical formulation of this task. Unlike these efforts, our work concentrates on the formalization of the data-generating process and the theoretical understanding. Thus, these two lines complement each other.

**Latent variable identification.** Identifying latent variables under nonlinear transformations is central to representation learning on complex unstructured data. Khemakhem et al. [35, 36], Hyvarinen and Morioka [37], Hyvarinen et al. [38] assume the availability of auxiliary information (e.g., domain/class labels) and that the latent variables' probability density functions have sufficiently different derivatives over domains/classes. However, many important concepts (e.g., object classes) are inherently discrete. Since latent variables are not equipped with differentiable density functions, identifying these concepts necessitates novel techniques. Our theory requires neither domain/class labels nor differentiable density functions and can accommodate discrete variables readily. Another line of studies [39, 40] refrains from the auxiliary information by making sparsity and mechanistic independence assumptions over latent variables, disregarding causal structures among the latent variables. Moreover, images may comprise abstract concepts and convey sophisticated interplay among concepts at various levels of abstraction. In this work, we address these limitations by formulating the concept space as a discrete hierarchical causal model, capturing concepts at distinct levels and their causal relations.

**Latent diffusion understanding.** Diffusion probabilistic models [74, 75, 28, 76–78] have recently become the workhorse for state-of-the-art image generation. Diffusion models' empirical success sparked a plethora of efforts to probe into their empirical properties. Kwon et al. [79], Park et al. [62] discover that the UNet bottleneck representation exhibits highly structured semantic properties, traversing over which manipulates the generated image in a meaningful manner. Choi et al. [80], Daras and Dimakis [81], Wu et al. [82], Sclocchi et al. [83] realize that early/late diffusion steps at the inference correlate with coarse/fine features in the output. Recently, Gandikota et al. [65] showcase that concepts are encoded by low-rank influences in latent diffusion models. The theoretical insights in our work consolidate these apparently separate strands of empirical observations and also lead to new understandings that could enhance empirical methodologies.

## A2 Proof for Theorem 4.4

**Condition 4.3** (Discrete Components Identification).

   *i [Connected Spaces] The continuous support $\mathcal{C} \subset \mathbb{R}^{n_c}$ is closed and connected.*

   *ii [Invertibility & Continuity]: The generating function $g$ in equation 1 is invertible, and for any fixed $\mathbf{d}$, $g(\mathbf{d}, \cdot)$ and its inverse are continuous.*

   *iii [Non-Subset Observed Children]: For any pair $d_i$ and $d_j$, one's observed children are not the subset of the other's, $Ch_\Gamma(d_i) \not\subset Ch_\Gamma(d_j)$.*

**Theorem 4.4** (Discrete Component Identification). *Under the generating process in Equation 1 and Condition 4.3-ii, the estimated discrete variable $\hat{\mathbf{d}}$ and the true discrete variable $\mathbf{d}$ are equivalent up to an invertible function, i.e., $\hat{\mathbf{d}} = h(\mathbf{d})$ with $h(\cdot)$ invertible. Moreover, if Condition 4.1 and Condition 4.3-iii further hold, we attain component-wise identifiability (Definition 4.2) and the bipartite graph $\Gamma$ up to permutation of component indices.*

*Proof of Theorem 4.4 Part 1.* The estimate $\hat{\mathbf{d}}$ and the true variable $\mathbf{d}$ are related through the map $[\hat{\mathbf{d}}, \hat{\mathbf{c}}] = \hat{g}^{-1} \circ g(\mathbf{d}, \mathbf{c})$. In the following, we show that the induced relation between $\mathbf{d}$ and $\hat{\mathbf{d}}$ is invertible under Condition 4.3-ii. The estimated generating process respects the conditions on the true generating process.

We denote that support of the estimate $\hat{\mathbf{d}}$ as $\hat{\Omega}^{(d)}$. First, we show by contradiction that for each state $k \in \Omega^{(d)}$, $k$ corresponds to at most one state $\hat{k} \in \hat{\Omega}^{(d)}$ of the estimate $\hat{\mathbf{d}}$.

Suppose that $k$ corresponds to two distinct states $\hat{k}_1$ and $\hat{k}_2$. That is, there exist $\mathbf{c}_1, \mathbf{c}_2 \in \mathcal{C}$ and $\hat{\mathbf{c}}_1, \hat{\mathbf{c}}_2 \in \hat{\mathcal{C}}$, such that $\hat{g}^{-1} \circ g(k, \mathbf{c}_1) = [\hat{k}_1, \hat{\mathbf{c}}_1]$ and $\hat{g}^{-1} \circ g(k, \mathbf{c}_2) = [\hat{k}_2, \hat{\mathbf{c}}_2]$. On one hand, As $g(k, \cdot)$ is a continuous function and $\mathcal{C}$ is connected, the image $\mathcal{I}(k) := g(k, \mathcal{C})$ is a connected set. On the other hand, $\hat{\mathcal{I}}(k_1) := \hat{g}(\hat{k}_1, \hat{\mathcal{C}})$ and $\hat{\mathcal{I}}(k_2) := \hat{g}(\hat{k}_2, \hat{\mathcal{C}})$ are two separate sets due to the invertibility and continuity of $\hat{g}$ and the closed-ness of $\hat{\mathcal{C}}$. To see this, invertibility implies that $\hat{\mathcal{I}}(k_1)$ and $\hat{\mathcal{I}}(k_2)$ are disjoint. The fact that $\hat{g}$ is continuous over $\hat{\mathbf{c}}$ and has a continuous inverse over $\hat{\mathbf{c}}$ implies that $\hat{\mathcal{I}}(k_1)$ and $\hat{\mathcal{I}}(k_2)$ preserve the closed-ness of $\hat{\mathcal{C}}$. The space formed by two disjoint closed subspaces is disconnected. Since $k$ corresponds to $\hat{k}_1$ and $\hat{k}_2$, it follows that $\mathcal{I}(k) = \hat{\mathcal{I}}_1 \cup \hat{\mathcal{I}}_2$ where $\hat{\mathcal{I}}_1$ and $\hat{\mathcal{I}}_2$ are nonempty subsets of $\hat{\mathcal{I}}_{\hat{k}_1}$ and $\hat{\mathcal{I}}_{\hat{k}_2}$ respectively and inherit their separability. As $\mathcal{I}(k)$ is a union of two nonempty separated sets, it is disconnected. This contradicts the connectedness of $\mathcal{I}(k)$. Therefore, for each state $k \in \Omega^{(d)}$, $k$ corresponds to at most one state $\hat{k} \in \hat{\Omega}^{(d)}$.

Having established that each state of $\mathbf{d}$ corresponds to at most one state of $\hat{\mathbf{d}}$, we now show that states $\hat{k}_1, \hat{k}_2$ of $\hat{\mathbf{d}}$ corresponding to distinct states $k_1 \neq k_2$ of $\mathbf{d}$ must also be distinct, i.e., $\hat{k}_1 \neq \hat{k}_2$ if $k_1 \neq k_2$. Suppose that $\exists k_1 \neq k_2$, such that the corresponding states $\hat{k}_1 = \hat{k}_2$. We denote $\hat{k} := \hat{k}_1 = \hat{k}_2$ and two arbitrary points $\mathbf{x}_1 := g(k_1, \mathbf{c}_1)$ and $\mathbf{x}_2 := g(k_2, \mathbf{c}_2)$ from modes $k_1$ and $k_2$ respectively. As the two estimated discrete states collapse at $\hat{k}$, it follows that

$$\mathbf{x}_1 = g(k_1, \mathbf{c}_1) = \hat{g}(\hat{k}, \hat{\mathbf{c}}_1) \tag{3}$$

$$\mathbf{x}_2 = g(k_2, \mathbf{c}_2) = \hat{g}(\hat{k}, \hat{\mathbf{c}}_2). \tag{4}$$

Since $\hat{g}(\hat{k}, \cdot)$ is continuous and $\hat{\mathcal{C}}$ is a connected set, the image $\hat{g}(\hat{k}, \hat{\mathcal{C}})$ is path-connected. Thus, we can find a path $f : [0, 1] \to \mathcal{X}$ such that $f(0) = \mathbf{x}_1$ and $f(1) = \mathbf{x}_2$. Also, each point on the path

$f$ has a positive probability density due to positive $\hat{p}(\hat{\mathbf{c}})$ and $\mathbb{P}\left(\hat{\mathbf{d}} = \hat{k}\right)$. However, the two images $g(k_1, \mathcal{C})$ and $g(k_2, \mathcal{C})$ are disconnected due to the invertibility of $g$ and $k_1 \neq k_2$. On any path from $\mathbf{x}_1$ to $\mathbf{x}_2$, there exists points $\mathbf{x}_0$ such that the density is strictly zero due to the discrete structure of $\mathbf{d}$. Thus, we have arrived at a contradiction. We have shown that if $k_1 \neq k_2$, the corresponding estimated states are distinct $\hat{k}_1 \neq \hat{k}_2$.

Since for for each $k \in \Omega^{(d)}$, $k$ corresponds to at most one state $\hat{k}$ and distinct states $k_1$, $k_2$ give rise to distinct states $\hat{k}_1$, $\hat{k}_2$, we have proven that for each $k \in \Omega^{(d)}$, $k$ corresponds to exactly one estimated state $\hat{k} \in \hat{\Omega}^{(d)}$.

$\square$

**Definition A2.1** (Mixture Oracles). Let $\mathbf{x}$ be a set of observed variables and $\mathbf{d} \in \Omega^{(d)}$ be a discrete latent variable. The mixture model is defined as $\mathbb{P}(\mathbf{x}) = \sum_{k \in \Omega^{(d)}} \mathbb{P}(\mathbf{d} = k)\,\mathbb{P}(\mathbf{x}|\mathbf{d} = k)$. A mixture oracle MixOracle($\mathbf{x}$) takes $\mathbb{P}(\mathbf{x})$ as input and returns the number of components $\left|\Omega^{(d)}\right|$, the weights $\mathbb{P}(\mathbf{d} = k)$ and the component $\mathbb{P}(\mathbf{x}|\mathbf{d} = k)$ for $k \in \Omega^{(d)}$. [3]

**Theorem A2** (Kivva et al. [24]). *Under Condition 4.1 and Condition 4.3-iii, on can reconstruct the bipartite graph $\Gamma$ between $\mathbf{d}$ and $\mathbf{x}$, and the joint distribution $\mathbb{P}(d_1 = k_1, \ldots, d_{n_d} = k_{n_d})$ from $\mathbb{P}(\mathbf{x})$ and MixOracle($\mathbf{x}$).*

*Proof of Theorem 4.4 Part 2.* **Step 1**: Given the first result in Theorem 4.4, we can identify the discrete state index $k$ for each realization of $\mathbf{x}$ (up to permutations). Since we can do this to all realizations of $\mathbf{x}$ and we are given $\mathbb{P}(\mathbf{x})$, we can compute the cardinality of the discrete subspace $|\Omega^{(d)}|$, the marginal distribution of each latent state $\mathbb{P}(\mathbf{d} = k)$, and the conditional distribution $\mathbb{P}(\mathbf{x}|\mathbf{d} = k)$ for $k \in \Omega^{(d)}$.

**Step 2**: Step 1 shows the availability of the mixture oracle MixOracle (i.e., $|\Omega|$, $\mathbb{P}(\mathbf{d} = k)$, and $\mathbb{P}(\mathbf{x}|\mathbf{d} = k)$ ) as defined in Definition A2.1. Now, all conditions employed in Theorem A2 are ready, namely Condition 4.1, Condition 4.3 iii, and MixOracle (the consequence of step 1). The derivation in Kivva et al. [24] entails identifying a map from the discrete subspace state index $\mathbf{d} = k$ where $k \in \Omega^{(d)}$ to all discrete components' state indices $[d_1, \ldots, d_{n_d}] = [k_1, \ldots, k_{n_d}]$ where $k_i \in \Omega_i^{(d)}$ is the state index of the $i$-th component $d_i$. Thus, we can utilize this map to identify the state index for each individual discrete variable $d_i$ from the global index $k$.

**Step 3**: As stated in Step 2, all conditions in Theorem A2 hold in our problem. Since Theorem Theorem A2 additionally identifies the bipartite graph $\Gamma$ between $\{x_1, x_2, x_3, \ldots\}$ and $\{d_1, d_2, d_3, \ldots\}$, the same follows in our case.

$\square$

## A3 Proof for Theorem 4.12

In this section, we present a proof for Theorem 4.12. Since all variables are discrete for this proof, for a set of variables $\mathbf{A}$, we adopt the notation $\mathbf{A} = i$ to indicate the joint state of all variables in $\mathbf{A}$.

As outlined in Section 4, we will derive Theorem 4.8 which serves as the bridge between the distributional information and the graphical information, equivalent to the role of Theorem A2 Sullivant et al. [49] in Dong et al. [20], Huang et al. [19].

To familiarize the reader with the context, we introduce Theorem A2 and the involved graphical definitions treks 4.6, t-separation 4.7, and its connection between d-separation [84].

**Definition 4.6** (Treks). A trek $T_{i,j}$ in a DAG from vertex $i$ to $j$ consists of a directed path $P_{ki}$ from $k$ to $i$ and a direct path $P_{kj}$ from $k$ to $j$, where we refer to $P_{ki}$ as the $i$ side and $P_{kj}$ as the $j$ side.

Intuitively, a trek is a path containing at most one fork structure and no collider structures. Given this definition, a notion of t-separation is introduced [51], reminiscent of the classic d-separation.

---

[3]We abuse the notation $\mathbb{P}(\cdot)$ to denote probability density functions for continuous variables and mass functions for discrete variables.

**Definition 4.7** (T-separation). Let $\mathbf{A}$, $\mathbf{B}$, $\mathbf{C_A}$, and $\mathbf{C_B}$ be subsets (not necessarily disjoint) of vertices in a DAG. Then $(\mathbf{C_A}, \mathbf{C_B})$ t-separates $\mathbf{A}$ and $\mathbf{B}$ if every trek from $\mathbf{A}$ to $\mathbf{B}$ passes through either a vertex in $\mathbf{C_A}$ on the $\mathbf{A}$ side of the trek or a vertex $\mathbf{C_B}$ on the $\mathbf{B}$ side of the trek.

**Theorem A1** (Equivalence between d-separation and t-separation [85]). *Suppose we have disjoint vertex sets $\mathbf{A}$, $\mathbf{B}$, and $\mathbf{C}$ in a DAG. Set $\mathbf{C}$ d-separates set $\mathbf{A}$ and set $\mathbf{B}$ if and only if there exists a partition $\mathbf{C} := \mathbf{C_A} \cup \mathbf{C_B}$ such that $(\mathbf{C_A}, \mathbf{C_B})$ t-separates $\mathbf{A} \cup \mathbf{C}$ and $\mathbf{B} \cup \mathbf{C}$.*

Theorem A1 shows that one can reformulate d-separation with a special form of t-separation. Thus, t-separation is at least as informative as d-separation. Further, as detailed in Dong et al. [20], t-separation can provide more information when latent variables are involved, benefiting from Theorem A2 [49].

**Theorem A2** (Covariance Matrices and Graph Structures [49]). *Given two sets of variables $\mathbf{A}$ and $\mathbf{B}$ from a linear model with graph $\mathcal{G}$, it follows that $rank(\Sigma_{\mathbf{A},\mathbf{B}}) = \min\{|\mathbf{L}| : \mathbf{L}$ t-separates $\mathbf{A}$ from $\mathbf{B}$ in $\mathcal{G}\}$, where $\Sigma_{\mathbf{A},\mathbf{B}}$ denotes the generic covariance matrix between $\mathbf{A}$ and $\mathbf{B}$.*

Theorem A2 reveals that one can access local latent graph structures, i.e., the cardinality of the minimal separation set between two subsets of observed variables, through computable statistical quantities, e.g., covariance matrix ranks. Dong et al. [20] utilize these local latent graph structures, together with graphical conditions, to develop their identification theory for linear hierarchical models. Ideally, if we can access such local latent structures in the discrete hierarchical model, we can apply the same graph search procedure and theorems in Dong et al. [20] to identify the discrete model. Nevertheless, Theorem A2 relies on the linearity of the causal model (i.e., each causal edge represents a linear function), which doesn't hold in the discrete case. This motivates us to derive a counterpart of Theorem A2 for discrete causal models.

To this end, we introduce a classic theorem (Theorem A3) that connects the non-negative rank of a joint probability table with latent variable states.

**Definition 4.5** (Non-negative Rank). The non-negative rank of non-negative $A \in \mathbb{R}_+^{m \times n}$ is equal to the smallest $p$ for which there exist $\mathbf{B} \in \mathbb{R}_+^{m \times p}$ and $\mathbf{C} \in \mathbb{R}_+^{p \times n}$ such that $\mathbf{A} = \mathbf{BC}$.

**Theorem A3** (Non-negative Rank and Probability Matrix Decomposition [50]). *Let $\mathbf{P} \in \mathbb{R}^{m \times n}$ be a bi-variate probability matrix. Then its non-negative rank $rank_+(\mathbf{P})$ is the smallest non-negative integer $p$ such that $\mathbf{P}$ can be expressed as a convex combination of $p$ rank-one bi-variate probability matrices.*

Given this machinery, we now derive Theorem 4.8 which provides equivalent information in discrete models as Theorem A2 in linear models.

**Theorem 4.8** (Implication of Rank Information on Latent Discrete Graphs). *Given two sets of variables $\mathbf{A}$ and $\mathbf{B}$ from a non-degenerate, faithful (Condition 4.1-i, Condition 4.10-i) discrete model $\mathcal{G}$, it follows that $rank_+(\mathbf{P_{A,B}}) = \min\{|Supp(\mathbf{L})| : a\ partition\ (\mathbf{L}_1, \mathbf{L}_2)\ t\text{-separates}\ \mathbf{A}\ and\ \mathbf{B}\ in\ \mathcal{G}\}$.*

*Proof.* We express the joint distribution table $\mathbf{P_{A,B}}$ as

$$\mathbf{P}(\mathbf{A} = i, \mathbf{B} = j) = \sum_{r \in [R]} \mathbf{P}(\mathbf{A} = i|\mathbf{L} = r)\mathbf{P}(\mathbf{B} = j|\mathbf{L} = r)\mathbf{P}(\mathbf{L} = r), \tag{5}$$

where $R \in \mathbb{N}^+$ is the smallest possible value. This is always possible since we can assign $\mathbf{L}$ as either $\mathbf{A}$ or $\mathbf{B}$ and obtain a trivial expression.

We note that $\mathbf{A} \setminus \mathbf{L}$, $\mathbf{B} \setminus \mathbf{L}$, and $\mathbf{L}$ are disjoint because if $\mathbf{A} \cap \mathbf{B}$ is nonempty, it must be a subset of $\mathbf{L}$. Since the graph $\mathcal{G}$ is non-degenerate (Condition 4.1-i) and faithful (Condition 4.10-i), Equation 5 implies the graphical condition that $\mathbf{A} \setminus \mathbf{L}$ and $\mathbf{B} \setminus \mathbf{L}$ are d-separate given $\mathbf{L}$.

The equivalence relation in Theorem A1 implies that a partition of $\mathbf{L}$ t-separates $\mathbf{A}$ and $\mathbf{B}$. Thus, the minimal cardinality $R$ is equal to the smallest number of discrete states of $\mathbf{L}$ that t-separates $\mathbf{A}$ and $\mathbf{B}$. Moreover, Theorem A3 implies that the minimal number of states is equal to the non-negative rank of $\mathbf{P_{A,B}}$, i.e., $R = rank^+(\mathbf{P_{A,B}})$, which concludes our proof. $\square$

With Theorem 4.8 in hand, we leverage existing structural identification results on *linear* hierarchical models (Theorem A10) to obtain the identification results of desire (Theorem 4.12).

We introduce formal definitions of linear models, pure children, and the minimal graph operator, which we refer to in the main text.

**Definition A3.4** (Linear Causal Models [20, 19]). A linear causal model is a DAG with variable set $\mathbf{V}$ and an edge set $\mathbf{E}$, where each causal variables $v$ is generated by its parents $\mathrm{Pa}(v)$ through a linear function:

$$v_i := \sum_{v_j \in \mathrm{Pa}(v)} a_{i,j} v_j + \epsilon_i, \tag{6}$$

where $a_{i,j}$ is the causal strength and $\epsilon_i$ is the exogenous variable associated with $v_i$.

**Definition A3.5** (Pure Children). A variable set $\mathbf{Y}$ are pure children of variables $\mathbf{X}$ in graph $\mathcal{G}$, iff $\mathrm{Pa}_\mathcal{G}(\mathbf{Y}) = \cup_{\mathbf{Y}_i \in \mathbf{Y}} \mathrm{Pa}_\mathcal{G}(\mathbf{Y}_i) = \mathbf{X}$ and $\mathbf{X} \cap \mathbf{Y} = \emptyset$. We denote the pure children of $\mathbf{X}$ in $\mathcal{G}$ by $\mathrm{PCh}_\mathcal{G}(\mathbf{X})$.

Basically, the definition dictates that variable $\mathbf{Y}$ has no other parents than $\mathbf{X}$.

**Definition 4.11** (Minimal-graph Operator [19, 20]). We can merge atomic covers $\mathbf{L}$ into $\mathbf{P}$ in $\mathcal{G}$ if (i) $\mathbf{L}$ is a pure child of $\mathbf{P}$, (ii) all elements of $L$ and $P$ are latent and $|\mathrm{Supp}(\mathbf{L})| = |\mathrm{Supp}(\mathbf{P})|$, and (iii) the pure children of $\mathbf{L}$ form a single atomic cover, or the siblings of $\mathbf{L}$ form a single atomic cover. We denote such an operator as the minimal-graph operator $\mathcal{O}_{\min}(\mathcal{G})$.

This operator merges certain structural redundancies not detectable from rank information [19, 20] (Lemma A9). Please refer to Figure A1 for an example.

**Definition A3.6** (Atomic Covers (Linear Models)). Let $\mathbf{A}$ be a set of variables in $\mathcal{G}$ with $|\mathbf{A}| = k$, where $t$ of the $k$ variables are observed variables, and the remaining $k - t$ are latent variables. $\mathbf{A}$ is an atomic cover if $\mathbf{A}$ contains a single observed variable, or if the following conditions hold:

(i) There exists a set of atomic covers $\mathcal{C}$, with $|\mathcal{C}| \geq k + 1 - t$, such that $\cup_{\mathbf{C} \in \mathcal{C}} \mathbf{C} \subseteq \mathrm{PCh}_\mathcal{G}(\mathbf{V})$ and $\forall \mathbf{C_1}, \mathbf{C_2} \in \mathcal{C}, \mathbf{C_1} \cap \mathbf{C_2} = \emptyset$.

(ii) There exists a set of covers $\mathcal{N}$, with $|\mathcal{N}| \geq k + 1 - t$, such that every element in $\cup_{\mathbf{N} \in \mathcal{N}} \mathbf{N}$ is a neighbour of $\mathbf{V}$ and $(\cup_{\mathbf{N} \in \mathcal{N}} \mathbf{N}) \cap (\cup_{\mathbf{C} \in \mathcal{C}} \mathbf{C}) = \emptyset$.

(iii) There does not exist a partition of $\mathbf{A} = \mathbf{A_1} \cup \mathbf{A_2}$ such that both $\mathbf{A_1}$ and $\mathbf{A_2}$ are atomic covers.

**Theorem A7** (Linear Hierarchical Model Conditions).

   *i [Rank Faithfulness]: All the rank constraints on the covariance matrices are entailed by the DAG.*

   *ii [Basic Graphical Conditions] For any $\mathsf{L} \in \mathbf{V}$, $\mathsf{L}$ belongs to at least one atomic cover (Definition A3.6) in the linear model $\mathcal{G}$ (Definition A3.4) and no latent variable is involved in any triangle structure (i.e., three mutually adjacent variables).*

   *iii [Graphical Condition on Colliders] In a latent graph $\mathcal{G}$, if (i) there exists a set of variables $\mathbf{C}$ such that every variable in $\mathbf{C}$ is a collider of two atomic covers $\mathbf{L_1}$, $\mathbf{L_2}$, and denote by $\mathbf{A}$ the minimal set of variables that d-separates $\mathbf{L_1}$ from $\mathbf{L_2}$, (ii) there is a latent variable in $\mathbf{L_1}, \mathbf{L_2}, \mathbf{C}$ or $\mathbf{A}$, then we must have $|\mathbf{C}| + |\mathbf{A}| \geq |\mathbf{L_1}| + |\mathbf{L_2}|$.*

**Definition A3.8** (Skeleton Operator [19, 20]). Given an atomic cover $\mathbf{A}$ in a graph $\mathcal{G}$, for all $a \in \mathbf{A}$, $a$ is latent, and all $c \in \mathrm{PCh}(\mathbf{A})$, such that $a$ and $c$ are not adjacent, we can draw an edge from $a$ to $c$. We denote such an operator as skeleton operator $\mathcal{O}_s(\mathcal{G})$.

The skeleton operator introduces additional edges to fully connect atomic clusters [19, 20], which are indistinguishable from the rank information (Lemma A9). Please refer to Figure A1 for an example.

**Lemma A9** (Rank Invariance Huang et al. [19]). *The rank constraints are invariant with the minimal-graph operator and the skeleton operator; that is, $\mathcal{G}$ and $\mathcal{O}_s(\mathcal{O}_{\min}(\mathcal{G}))$ are rank equivalent.*

**Theorem A10** (Linear Hierarchical Model Identification [20]). *Suppose the $\mathcal{G}$ is a linear latent causal model (Definition A3.4) that satisfies Condition A7. Then the hierarchical causal model $\mathcal{G}$ is identifiable up to the Markov equivalent class of $\mathcal{O}_s(\mathcal{O}_{\min}(\mathcal{G}))$.*

We note that linear model conditions (Condition A7) and discrete model conditions (Condition 4.10) differ mainly in the substitutes of variables in the linear models with states in the discrete models. This originates from the local graph structures we can access, i.e., states in Theorem 4.8 and variables in Theorem A2. The skeleton operator $\mathcal{O}_{\min}$ (Definition A3.8 is not necessary under Condition 4.10 since each cover represents a discrete variable whose states must all be connected to its neighbors.

We now present Theorem 4.12 and its proof.

**Condition 4.10** (Discrete Hierarchical Model Conditions)**.**

    *i*  *[Faithfulness] All the conditional independence relations are entailed by the DAG.*

   *ii*  *[Basic Graphical Conditions] Each latent variable $z \in \mathbf{Z}$ corresponds to a unique atomic cover in $\mathcal{G}$ and no $z$ is involved in any triangle structure (i.e., three mutually adjacent variables).*

  *iii*  *[Graphical Condition on Colliders] In a latent graph $\mathcal{G}$, if (i) there exists a set of variables $\mathbf{C}$ such that every variable in $\mathbf{C}$ is a collider of two atomic covers $\mathbf{L_1}$, $\mathbf{L_2}$, and denote by $\mathbf{A}$ the minimal set of variables that d-separates $\mathbf{L_1}$ from $\mathbf{L_2}$, (ii) there is a latent variable in $\mathbf{L_1}, \mathbf{L_2}, \mathbf{C}$ or $\mathbf{A}$, then we must have $|Supp(\mathbf{C})| + |Supp(\mathbf{A})| \geq |Supp(\mathbf{L_1})| + |Supp(\mathbf{L_2})|$.*

**Theorem 4.12** (Discrete Hierarchical Identification)**.** *Suppose the causal model $\mathcal{G}$ satisfies Condition 4.1 and Condition 4.10 We can identify $\mathcal{G}$ up to the Markov equivalence class of $\mathcal{O}_{\min}(\mathcal{G})$.*

*Proof.* We observe that the linearity condition (Definition A3.4) in Theorem A10 is only utilized to invoke Theorem A2 to access the cardinality of the smallest t-separation set between any two sets of observed variables in the linear model. Through this, the graph identification results in Theorem A10 are derived based on a graph search algorithm repeatedly querying partial graph structures under Condition A7.

For discrete models (Condition 4.1), Theorem 4.8 supplies partial graph structures equivalent to Theorem A2. The difference is that Theorem A2 returns the number of variables in the smallest t-separation set while Theorem 4.8 returns the number of states in the smallest t-separation set. Thus, running Algorithm 1 up to Step 9 (i.e., the original search algorithm Dong et al. [20] with a different rank oracle in Theorem 4.8 highlighted in blue) will return a graph with latent nodes representing discrete states. Algorithm 1 is guaranteed to correctly discover all the atomic covers (Theorem A10) and each atomic cover corresponds to a latent discrete variable (Condition 4.10-ii). Thus, we can obtain each true latent variable by merging all the latent nodes $\mathbf{A}_L$ in each atomic cover $\mathbf{A}$ into a discrete latent variable $z$ whose support cardinality $|\mathrm{Supp}(z)|$ equals to the number of latent nodes $|\mathbf{A}_L|$. We highlight this procedure (Step 9 in Algorithm 1). Moreover, as all latent nodes (i.e., latent states) in an atomic cover belong to one discrete variable, these latent nodes in adjacent atomic covers must be fully connected. Thus, we do not need the skeleton operator $\mathcal{O}_s$ as for linear models (Theorem A10). This concludes our proof for Theorem 4.12.

$\square$

Theorem A12 follows the same reasoning as in Theorem 4.12, with the main difference in organizing latent nodes/states into latent discrete variables.

**Condition A3.11** (Discrete Hierarchical Model Conditions for Identical Supports)**.**

    *i*  *[Faithfulness]: All the conditional independence relations are entailed by the DAG.*

   *ii*  *[Basic Graphical Conditions]: Each latent variable $z \in \mathbf{Z}$ belongs to at least one atomic cover in $\mathcal{G}$ and no $z$ is involved in any triangle structure (i.e., three mutually adjacent variables).*

  *iii*  *[Graphical Condition on Colliders]: In a latent graph $\mathcal{G}$, if (i) there exists a set of variables $\mathbf{C}$ such that every variable in $\mathbf{C}$ is a collider of two atomic covers $\mathbf{L_1}$, $\mathbf{L_2}$, and denote by $\mathbf{A}$ the minimal set of variables that d-separates $\mathbf{L_1}$ from $\mathbf{L_2}$, (ii) there is a latent variable in $\mathbf{L_1}, \mathbf{L_2}, \mathbf{C}$ or $\mathbf{A}$, then we must have $|Supp(\mathbf{C})| + |Supp(\mathbf{A})| \geq |Supp(\mathbf{L_1})| + |Supp(\mathbf{L_2})|$.*

We introduce the skeleton operator $\mathcal{O}_s$ [19, 20] (Definition A3.8) that include edges between adjacent covers indistinguishable to rank information.

**Theorem A12** (Discrete Hierarchical Identification on Identical Supports)**.** *Suppose the causal model $\mathcal{G}$ satisfies Condition 4.1-i, Condition A3.11, and $|Supp(z)| = K \geq 2$ for all $z \in \mathbf{Z}$. We can identify $\mathcal{G}$ up to the Markov equivalence class of $\mathcal{O}_s(\mathcal{O}_{\min}(\mathcal{G}))$.*

*Proof.* The bulk of the proof overlaps with the proof of Theorem 4.12. Following the same reasoning of the proof of Theorem 4.12, we can obtain a graph with latent nodes representing discrete states before Step 9 and Step 10 in Algorithm 1. Under the identical support condition in Theorem A12, we

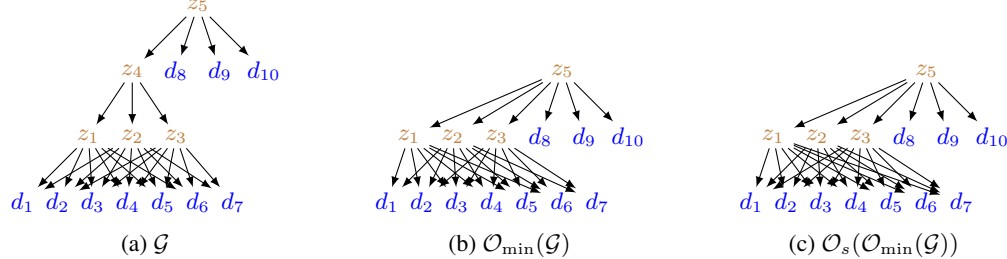

$$(a)\ \mathcal{G} \qquad\qquad (b)\ \mathcal{O}_{\min}(\mathcal{G}) \qquad\qquad (c)\ \mathcal{O}_s(\mathcal{O}_{\min}(\mathcal{G}))$$

Figure A1: **The discrete graph $\mathcal{G}$ satisfies conditions in Theorem A12 (i.e., identical supports).** After applying the minimal-graph operator to the graph $\mathcal{G}$, $z_4$ is merged to its parent $z_5$, and the rank constraints do not change. After applying the skeleton operator to the graph in (b), $z_1$ has an edge to $d_7$ and $z_3$ has an edge to $d_1$. We adopt this example from Huang et al. [19].

can directly group $K$ states in an atomic cover into a latent variable as in Algorithm 1-Step 10. Since the true latent variable cardinality is known to be identical, we don't need Condition 4.1-ii, iii to ensure the structure is well defined. Under Condition A3.11, each atomic cover may contain multiple discrete latent variables, depending on the cover size. It could be possible that one latent variable is not connected to all latent variables in an adjacent atomic cover, as in the linear model case. However, this difference cannot be detected from the rank information (Lemma A9). Thus, we need to retain the skeleton operator $\mathcal{O}_s$ inherited from Theorem A10 This concludes our proof for Theorem A12. $\quad\square$

---

**Algorithm 1: The overall procedure for Rank-based Discrete Latent Causal Model Discovery.** [20] We denote the latent nodes in an atomic cover $\mathbf{A}$ as $\mathbf{A}_L$ and all observed nodes in the model $\mathcal{G}$ as $\mathbf{X}_{\mathcal{G}}$. We use blue color to highlight our modifications needed for Theorem 4.12 and Theorem A12 respectively.

**Input** : Samples from all $n$ observed variables $\mathbf{X}_{\mathcal{G}}$
**Output**: Markov equivalence class $\mathcal{G}'$

1 **def** *LatentVariableCausalDiscovery($\mathbf{X}_{\mathcal{G}}$)*:
2      Phase 1: $\mathcal{G}' = \text{FindCISkeleton}(\mathbf{X}_{\mathcal{G}})$ (Algorithm 2);
3      **for** *Each $\mathcal{Q}$, a group of overlapping maximal cliques, in $\mathcal{G}'$* **do**
4          Set an empty graph $\mathcal{G}''$, $\mathbf{X}_{\mathcal{Q}} = \cup_{\mathbf{Q}\in\mathcal{Q}}\mathbf{Q}$, $\mathbf{N}_{\mathcal{Q}} = \{\mathsf{N}:\exists\mathsf{X}\in\mathbf{X}_{\mathcal{Q}}\ \text{s.t.}\ \mathsf{N},\mathsf{X}\ \text{are adjacent in}\ \mathcal{G}'\}$;
5          Phase 2: $\mathcal{G}'' = \text{FindCausalClusters}(\mathcal{G}'', \mathbf{X}_{\mathcal{Q}} \cup \mathbf{N}_{\mathcal{Q}})$ (Algorithm 3);
6          Phase 3: $\mathcal{G}'' = \text{RefineCausalClusters}(\mathcal{G}'', \mathbf{X}_{\mathcal{Q}} \cup \mathbf{N}_{\mathcal{Q}})$ (Algorithm 5);
7          Transfer the estimated DAG $\mathcal{G}''$ to the Markov equivalence class and update $\mathcal{G}'$ by $\mathcal{G}''$;
8      Orient remaining causal directions that can be inferred from v structures;
9      Theorem 4.12: replace each cover $\mathbf{A}$ to a discrete variable $z$ with $|\text{Supp}(z)| := |\mathbf{A}_L|$ & update $\mathcal{G}'$ ;
10      Theorem A12: convert each cover $\mathbf{A}$ to $\log_K(|\mathbf{A}_L|)$ discrete variables of cardinality $K$ & update $\mathcal{G}'$ ;
11      **return** $\mathcal{G}'$

---

## A4   Synthetic Data Experiments

**Data-generating processes.** For the hierarchical model $\mathcal{G}$, we randomly sample the parameters for each causal module, i.e., conditional distributions $p(z_i|\text{Parents}(z_i))$, according to a Dirichlet distribution over the states of $z_i$ with coefficient 1. For simplicity, we follow conditions in Theorem A12 and set the support size of latent variables to 2. Like Kivva et al. [24], we build the generating process from $\mathbf{d}$ to the observed variables $\mathbf{x}$ (i.e., graph $\Gamma$) by a Gaussian mixture model where each state of the discrete subspace corresponds to one component/mode in the mixture model. We truncate the support of each component to improve the invertibility (Condition 4.3-ii). The graphs are exhibited in Figure A2 and Figure A3.

**Metrics.** We adopt F1 score (i.e., $\frac{2\text{Precision}\cdot\text{Recall}}{\text{Precision}+\text{Recall}}$ ) to assess the graph learning results [20]. We compute recall and precision by checking whether the estimated model correctly retrieves edges in the true causal graph. Ranging between 0 to 1, high F1 scores indicate the search algorithm can recover ground-truth causal graphs. We repeat each experiment over at least 5 random seeds.

**Algorithm 2: Phase1: FindCISkeleton [20]** (Stage 1 of PC [51]). We denote the joint probability table between two sets $\mathbf{A}$ and $\mathbf{B}$ as $\mathbf{P_{A,B}}$, the adjacent nodes as Adj, the non-negative rank with $\text{rank}^+$, and the collection of d-separation sets as Sepset. We use blue color to highlight our modifications.

---

**Input** : Samples from observed variables $\mathbf{X}_{\mathcal{G}}$
**Output** : CI skeleton $\mathcal{G}'$

1 **def** *Stage1PC($\mathbf{X}_{\mathcal{G}}$)*:
2    Initialize a complete undirected graph $\mathcal{G}'$ on $\mathbf{X}_{\mathcal{G}}$;
3    **repeat**
4      **repeat**
5        Select an ordered pair $\mathsf{X}, \mathsf{Y}$ that are adjacent in $\mathcal{G}'$, s.t., $|\text{Adj}_{\mathcal{G}'}(\mathsf{X})\backslash\{\mathsf{Y}\}| \geq n$;
6        Select a subset $\mathbf{S} \subseteq \text{Adj}_{\mathcal{G}'}(\mathsf{X})\backslash\{\mathsf{Y}\}$ s.t., $|\mathbf{S}| = n$;
7        If $\text{rank}^+(\mathbf{P}_{\{\mathsf{X}\}\cup\mathbf{S},\{\mathsf{Y}\}\cup\mathbf{S}}) = |\mathbf{S}|$, delete the edge between $\mathsf{X}$ and $\mathsf{Y}$ from $\mathcal{G}'$ and record $\mathbf{S}$ in Sepset($\mathsf{X}, \mathsf{Y}$) and Sepset($\mathsf{Y}, \mathsf{X}$).;
8      **until** *all $\mathsf{X}, \mathsf{Y}$ s.t., $|Adj_{\mathcal{G}'}(\mathsf{X})\backslash\{\mathsf{Y}\}| \geq n$ and all $\mathbf{S} \subseteq Adj_{\mathcal{G}'}(\mathsf{X})\backslash\{\mathsf{Y}\}, |\mathbf{S}| = n$, tested.*;
9      n:=n+1;
10    **until** *no adjacent $\mathsf{X}, \mathsf{Y}$ s.t., $|Adj_{\mathcal{G}'}(\mathsf{X})\backslash\{\mathsf{Y}\}| < n$*;
11    **return** $\mathcal{G}'$

---

**Implementation details.** Our method comprises two stages: 1) learning the bottom-level discrete variable $\mathbf{d}$ and the bipartite graph $\Gamma$ from the observed variable $\mathbf{x}$; 2) learning the latent hierarchical model $\mathcal{G}$ given the bottom-level discrete variable $\mathbf{d}$ discovered in 1). For stage 1), we follow the clustering implementation in Kivva et al. [24] under the same hyper-parameter setup as in the original implementation. For stage 2), we apply Algorithm 1 to learn the hierarchical model $\mathcal{G}$. We opt for Step 10 in Algorithm 1 because we evaluate graphs with binary latent variables that meet the conditions of Theorem A12. Following Anandkumar et al. [52], Mazaheri et al. [53], we perform conventional rank computation rather than non-negative rank computation and find this replacement satisfactory. We conduct our experiments on a cluster of 64 CPUs. All experiments can be finished within half an hour. The search algorithm implementation is adapted from Dong et al. [20].

**Graphical structures.** Table 1 and Table 2 correspond to Figure A2 and Figure A3 respectively. As mentioned above, the graphs meet the conditions of Theorem A12 with the latent variable cardinality equal to two (binary variables).

## A5 Real-world Experiments

### A5.1 Implementation Details

We employ the pre-trained latent diffusion model [28] SD v1.4 across all our experiments. The inference process consists of 50 steps.

For experiments in Section 7.1, we inject concepts by appending keywords to the original prompt. For instance, we inject the concept pair ("sketch", "wide eyes") in Figure A5 as follows. For the inference steps $0-10$, we feed the text prompt "A picture of a person", for steps $10-20$, "a photo of a person, in a sketch style", and for steps $20-50$, "a photo of a person, in a sketch style, with wide eyes". For the reverse injection order (injecting "wide eyes" before "sketch"), we inject the following prompts at the three-step stages: "A picture of a person", "a photo of a person, with wide eyes", and "a photo of a person, with wide eyes, in a sketch style".

For experiments understanding the UNet's latent presentation (Figure 5), we adopt the open-sourced code of Park et al. [62].

For the attention sparsity experiment (Figure A4), we randomly generate images with the pre-train latent diffusion model and record their attention score across layers. To compute the relative sparsity, we select the threshold as $1/4096$ and compute the proportion of the attention scores over this threshold. For the attention visualization, we randomly select a head from the last attention module in the UNet architecture.

We follow the implementation of Gandikota et al. [65] to train concept sliders of various ranks. We adopt their evaluation protocol to obtain CLIP and LPIPS scores over 20 randomly sampled images for each rank, concept, and scale combination. We evaluate ranks in $\{2, 4, 8\}$ and scales $\{1, 2, 3, 4, 5\}$.

**Algorithm 3: Phase2: FindCausalClusters** [20]. We use $||\cdot||$ to denote the number of all elements in a set of sets. We use the term "nodes" to refer to dummy variables that represent states rather than causal variables in the intermediate graph. We use blue color to highlight our modifications.

---

**Input** : Samples from $n$ observed variables $\mathbf{X}_{\mathcal{G}}$
**Output** : Graph $\mathcal{G}'$

1 **def** *FindCausalClusters($\mathcal{G}'$, $\mathbf{X}_{\mathcal{G}}$)*:
2    Active set $\mathcal{S} \leftarrow \mathcal{X}_{\mathcal{G}} = \{\{\mathsf{X}_1\},...,\{\mathsf{X}_n\}\}, k \leftarrow 1$ ;        `// S is a set of covers`
3    **repeat**
4      $\mathcal{G}', \mathcal{S}$, found = Search($\mathcal{G}', \mathcal{S}, \mathbf{X}_{\mathcal{G}}, k$) ;        `// Only when nothing can be found`
5      If found $= 1$ then $k \leftarrow 1$ else $k \leftarrow k+1$ ;     `// udner current k do we add k by 1`
6    **until** $k$ *is sufficiently large*;
7    **return** $\mathcal{G}'$;
8 **def** *Search($\mathcal{G}', \mathcal{S}, \mathbf{X}_{\mathcal{G}}, k$)*:
9    Rank deficiency set $\mathbb{D} = \{\}$ ;        `// To store rank deficient combinations`
10    **for** $\mathcal{T} \in PowerSet(\mathcal{S})$ *(from $\mathcal{S}$ to $\emptyset$)* **do**
11      $\mathcal{S}' \leftarrow (\mathcal{S}\backslash\mathcal{T}) \cup (\cup_{\mathbf{T}\in\mathcal{T}}\text{PCh}_{\mathcal{G}'}(\mathbf{T}))$ ;        `// Unfold S to get S'`
12      **for** $t = k$ *to* $0$ **do**
13        **repeat**
14          Draw a set of $t$ observed covers $\mathcal{X} \subset \mathcal{S}' \cap \mathcal{X}_{\mathcal{G}}$;
15          **repeat**
16            Draw a set of covers $\mathcal{C} \subset \mathcal{S}'\backslash\mathcal{X}$, s.t., $||\mathcal{C}|| = k - t + 1$ and get $\mathcal{N} \leftarrow \mathcal{S}'\backslash(\mathcal{X} \cup \mathcal{C})$;
17            **if** $rank^+(\mathbf{P}_{\mathcal{C}\cup\mathcal{X},\mathcal{N}\cup\mathcal{X})} = k$ *and NoCollider($\mathcal{C}, \mathcal{X}, \mathcal{N}$)* **then** Add $\mathcal{C}$ to $\mathbb{D}$ ;
18          **until** *all $\mathcal{C}$ exhausted*;
19          **if** $\mathbb{D} \neq \emptyset$ **then**
20            **for** $\mathcal{D}_i \in \mathbb{D}$ **do**
21              **if** $|Pa_{\mathcal{G}'}(\mathcal{D}_i) \cup \mathbf{X}| = k$ **then** $\mathbf{P} \leftarrow \text{Pa}_{\mathcal{G}'}(\mathcal{D}_i) \cup \mathbf{X}$ ;
22              **else** Create new latent nodes $\mathbf{L}$, s.t., $\mathbf{P} \leftarrow \mathbf{L} \cup \text{Pa}_{\mathcal{G}'}(\mathcal{D}_i) \cup \mathbf{X}$ and $|\mathbf{L}| = k - |\text{Pa}_{\mathcal{G}'}(\mathcal{D}_i) \cup \mathbf{X}|$ ;
23              Update $\mathcal{G}'$ by taking elements of $\mathcal{D}_i$ as the pure children of $\mathbf{P}$;
24              **if** $\mathbf{P}$ *is atomic* **then** Update $\mathcal{S} \leftarrow (\mathcal{S}\backslash\mathcal{D}_i) \cup \mathbf{P}$ ;
25            **return** $\mathcal{G}', \mathcal{S}$, True ;        `// Return to search with k = 1`
26        **until** *all $\mathcal{X}$ exhausted*;
27    **return** $\mathcal{G}', \mathcal{S}$, False ;        `// Return to search with k ← k + 1`

---

**Algorithm 4: NoCollider** [20]. We use blue color to highlight our modifications.

---

**Input** : $\mathcal{C}, \mathcal{X}, \mathcal{N}$
**Output** : Whether there exists $\mathbf{O} \in \mathcal{C}$ s.t., $\mathbf{O}$ is a collider of $\mathcal{C}\backslash\{\mathbf{O}\}$ and $\mathcal{N}$

1 **def** *NoCollider($\mathcal{C}, \mathcal{X}, \mathcal{N}$)*:
2    **for** $c = 1$ *to* $|\mathcal{C}| - 1$ **do**
3      Draw $\mathcal{C}' \subset \mathcal{C}$ s.t., $|\mathcal{C}'| = c$;
4      **repeat**
5        **if** $rank^+(\mathbf{P}_{\mathcal{C}'\cup\mathcal{X},\mathcal{N}\cup\mathcal{X}}) < ||\mathcal{C}' \cup \mathcal{X}||$ **then** **return** False ;
6      **until** *all $\mathcal{C}'$ exhausted*;
7    **return** True

---

The rank selection technique, inspired by Ding et al. [67], involves multiplying each LoRA's inner dimension with a scalar parameter and imposing $\ell_0$ penalty on these scalar parameters. The weight on the $\ell_0$ penalty is selected from $\{1e - 1, 1e - 2, 1e - 3, 1e - 4, 1e - 5\}$. We repeat each run for at least three random seeds. The code can be found here.

We conduct all our experiments on 2 Nvidia L40 GPUs. Each image inference takes the same time as in standard SD v1.4 (i.e., within two minutes). Each concept slider in Figure A7 takes around half an hour to train.

### A5.2   Sparsity in the Hierarchical Model

To verify the sparse structure condition in Theorem 4.12, we view the attention sparsity in the LD model as an indicator of the connectivity between a specific hierarchical level and the bottom concept level. Figure A4 visualizes the attention sparsity of an LD model over diffusion steps and specific

---

**Algorithm 5:** Phase3: RefineCausalClusters

---

**Input** : Graph $\mathcal{G}'$
**Output** : Refined graph $\mathcal{G}'$

1 **def** *RefineCausalCLusters($\mathcal{G}'$, $\mathbf{X}_\mathcal{G}$)*:
2      **repeat**
3          Draw an atomic cover $\mathbf{V}$ from $\mathcal{G}'$;
4          Delete $\mathbf{V}$, neighbours of $\mathbf{V}$ that are latent, and all relating edges from $\mathcal{G}'$ to get $\hat{\mathcal{G}}$;
5          $\mathcal{G}' = \text{FindCausalClusters}(\hat{\mathcal{G}}, \mathbf{X}_\mathcal{G})$;
6      **until** *No more $\mathbf{V}$ found and all $\mathbf{V}$ exhausted*;
7      **return** $\mathcal{G}'$

---

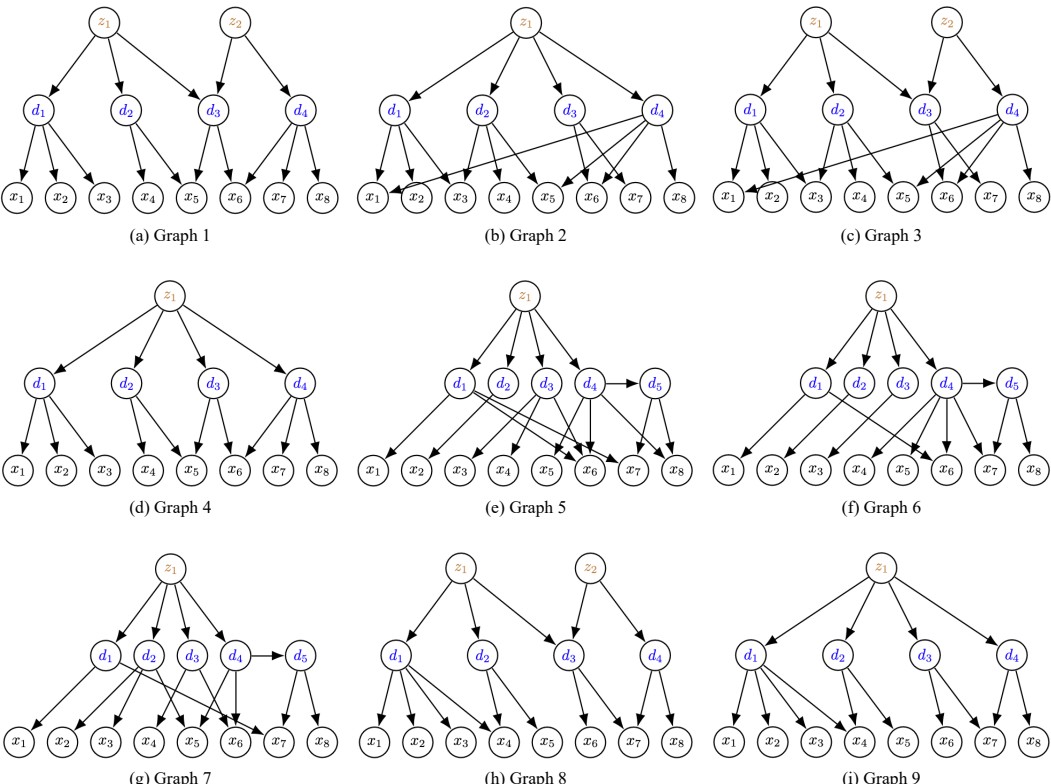

Figure A2: **Causal graphs evaluated in Table 1.** We denote the observed variables with $x$, the bottom-level latent discrete variables with $d$, and the high-level latent discrete variables with $z$.

attention patterns in the model. We observe that the sparsity increases as the generative process progresses, which reflects that the connectivity between the hierarchical level ($\mathbf{z}_{d,\mathcal{S}_t}$) and the bottom level variable ($\mathbf{d}$) becomes sparse and more local as we march down the hierarchical structure, which indicates a gradual localization of the concept.

### A5.3 Discovering Hierarchical Orders from Diffusion Models

We provide further evidence that latent representations at different diffusion steps correspond to different levels of the hierarchical causal model. We select concept pairs, each with higher-level and lower-level concepts. For example, in ("sketch," "wide eyes"), "sketch" is more global, while "wide eyes" is more local. We alter the text prompt during diffusion generation for concept injection, appending "in a sketch style" to inject "sketch" (see Appendix A5 for prompts). In Figure A5, global concepts are successfully injected at early diffusion steps and local ones at late steps (top row). Reversing this order fails, as shown in the bottom row. For example, injecting "sketch" early and "wide eyes" late renders both correctly, but the global concept "sketch" is absent under the reverse

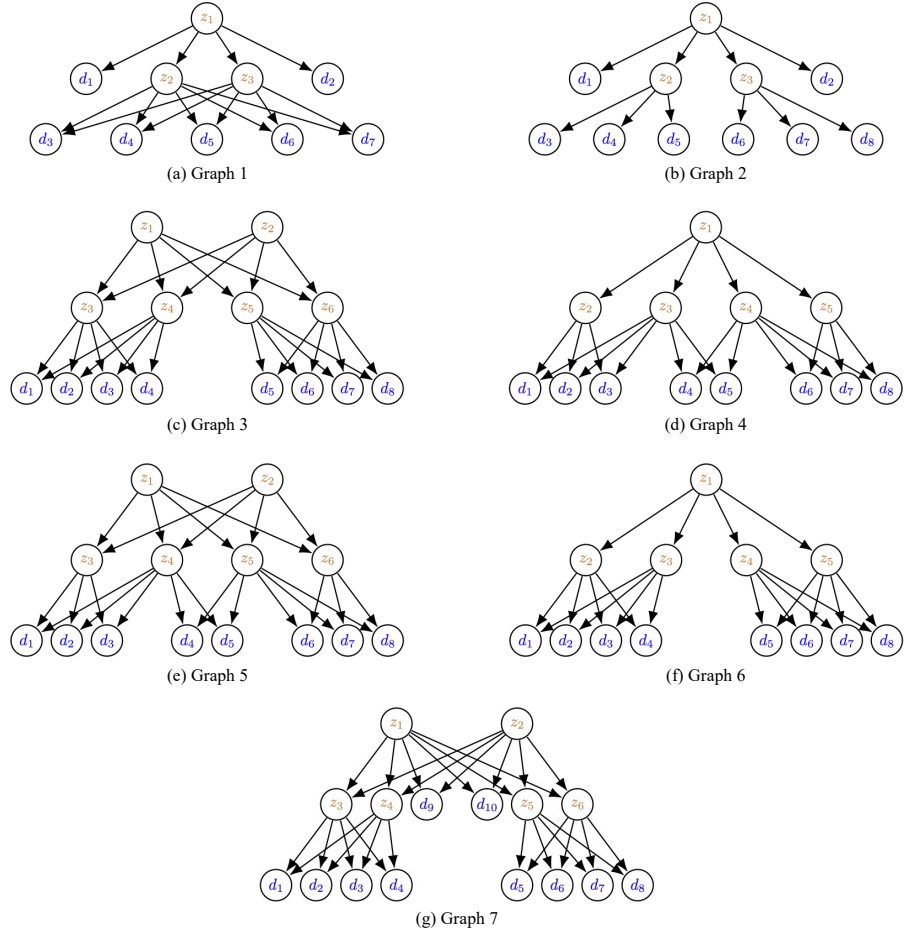

Figure A3: **Causal graphs evaluated in Table 2.** We denote the bottom-level latent discrete variables with $d$, and high-level latent discrete variables with $z$. Since baselines cannot extract discrete subspaces, we directly feed the algorithms the bottom-level discrete variables and test their structure learning performances.

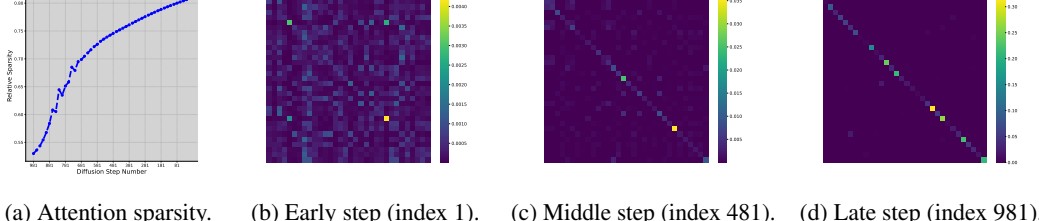

(a) Attention sparsity.  (b) Early step (index 1).  (c) Middle step (index 481).  (d) Late step (index 981).

Figure A4: **Sparsity patterns in latent diffusion models' attention.** We compute the proportions of the attention scores lower than a fixed threshold over the entire model. We can observe that the sparsity increases greatly towards small timesteps, i.e., the lower levels of the hierarchical model, which verifies our theory.

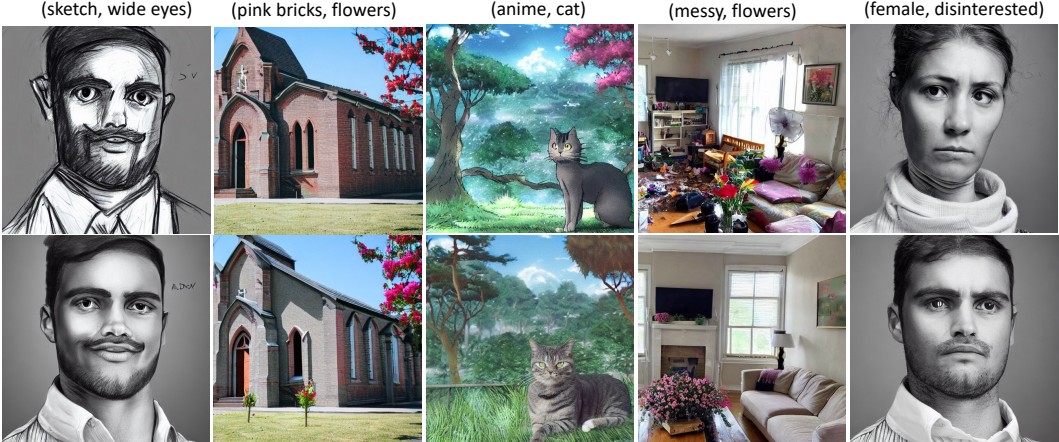

Figure A5: **Hierarchical Concept Ordering.** We inject concepts of distinct abstraction levels into the generating process at different time steps. In the top row, the concept injection follows the hierarchical order, which renders injected concepts faithfully. The bottom row reverses the hierarchical order and cannot incorporate concepts properly. More examples in Figure A8.

injection order. This supports our theory that concepts are hierarchically organized, with higher-level concepts related to earlier diffusion steps.

## A5.4 Causal Sparsity for Concept Extraction

Figure A6 shows that indeed concepts at different abstraction levels have desirable representations at different ranks. For instance, the concept of bright weather is appropriately conveyed by a rank-2 LoRA and higher-rank LoRAs alter the background. The same observation occurs to other concepts, where inadequate ranks fail to capture the concept faithfully and unnecessary ranks entangle the target concept with other attributes.

Figure A7 presents the CLIP and LPIPS evaluation for the baseline and our approach, where the CLIP score evaluates the alignment between the image and the target description and the LPIPS score measures the structure change between the edited image and the original image. We can observe that under the sparsity constraint, our approach attains the highest CLIP score and the lowest LPIPS score when compared with the baselines of several ranks, indicating a higher level of alignment and a lower level of undesirable entanglement.

## A5.5 More Examples

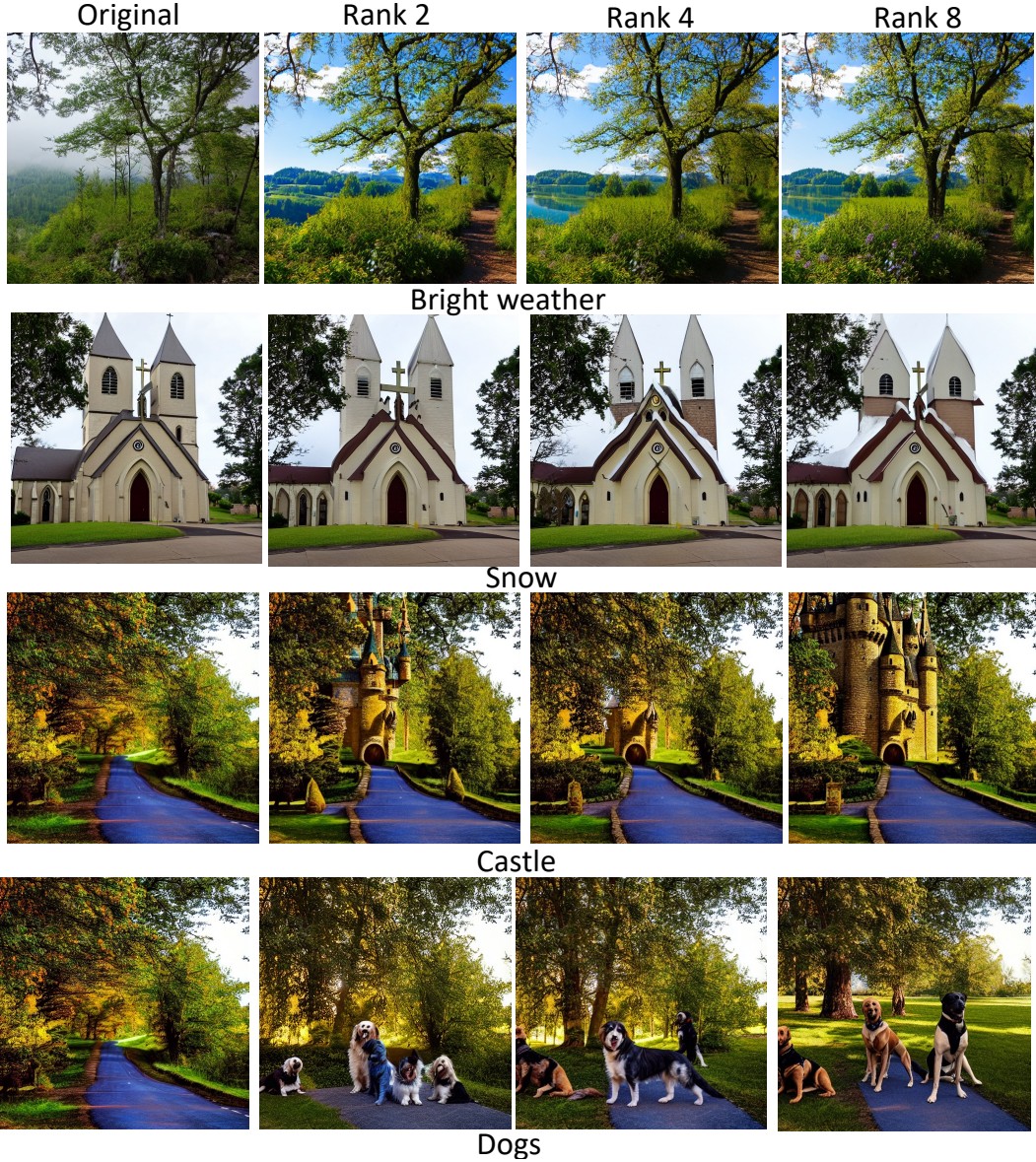

Figure A6: **Concepts have varying levels of sparsity.** We show that concepts of various abstraction levels correspond to different sparsity levels. For instance, bright weather is appropriately conveyed by a rank-2 LoRA and higher-rank LoRAs alter the background. Inadequate ranks fail to capture the concept faithfully and unnecessary ranks entangle the target concept with other attributes.

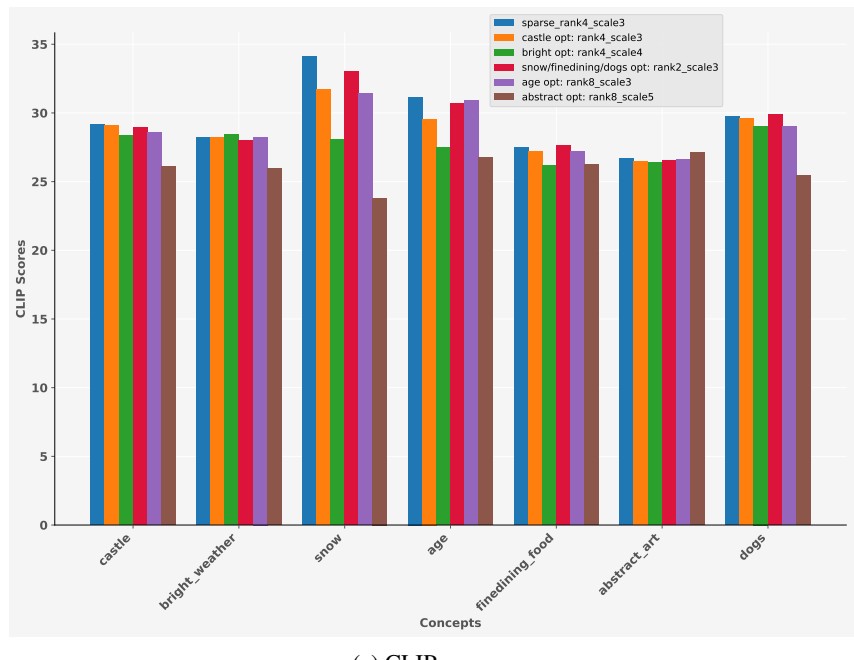

(a) CLIP scores.

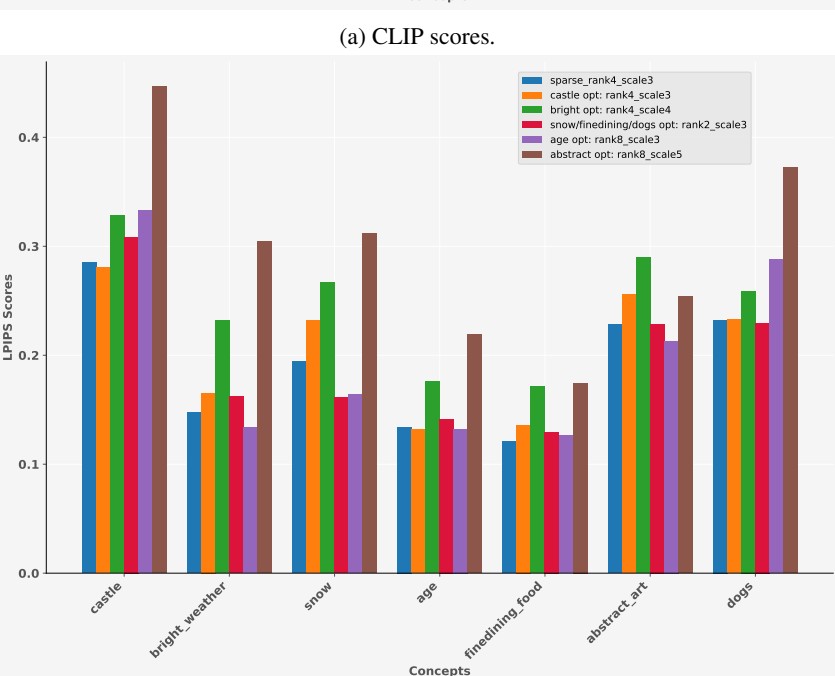

(b) LPIPS scores.

Figure A7: **CLIP/LPIPS evaluation.** We evaluate our approach and baselines at individual rank constraints. A high CLIP score is favorable as it indicates semantic alignment. A low LPIPS score is more favorable as it indicates minimal excessive changes. We compare our method "sparse" with the optimal fixed rank setting on each concept. For instance, "castle opt: rank4_scale3" indicates that the optimal setting for the concept "castle" is the LoRA of rank 4 and scale 3. With a adaptive rank selection, our approach outperforms or keeps up with the optimal fixed setting across different concepts. We repeat each training over three random seeds.

Cyberpunk · High   Church style · With tree   Young · White hair   Crowd people · Excavator   Kitchen · With flowers

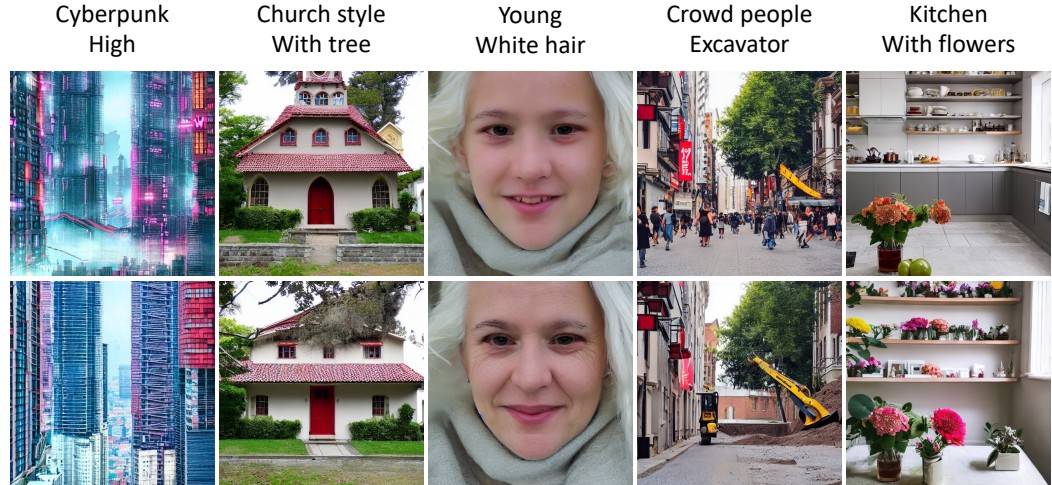

Figure A8: **More examples for Figure A5**.

Original   T   0.6T   Original   T   0.6T

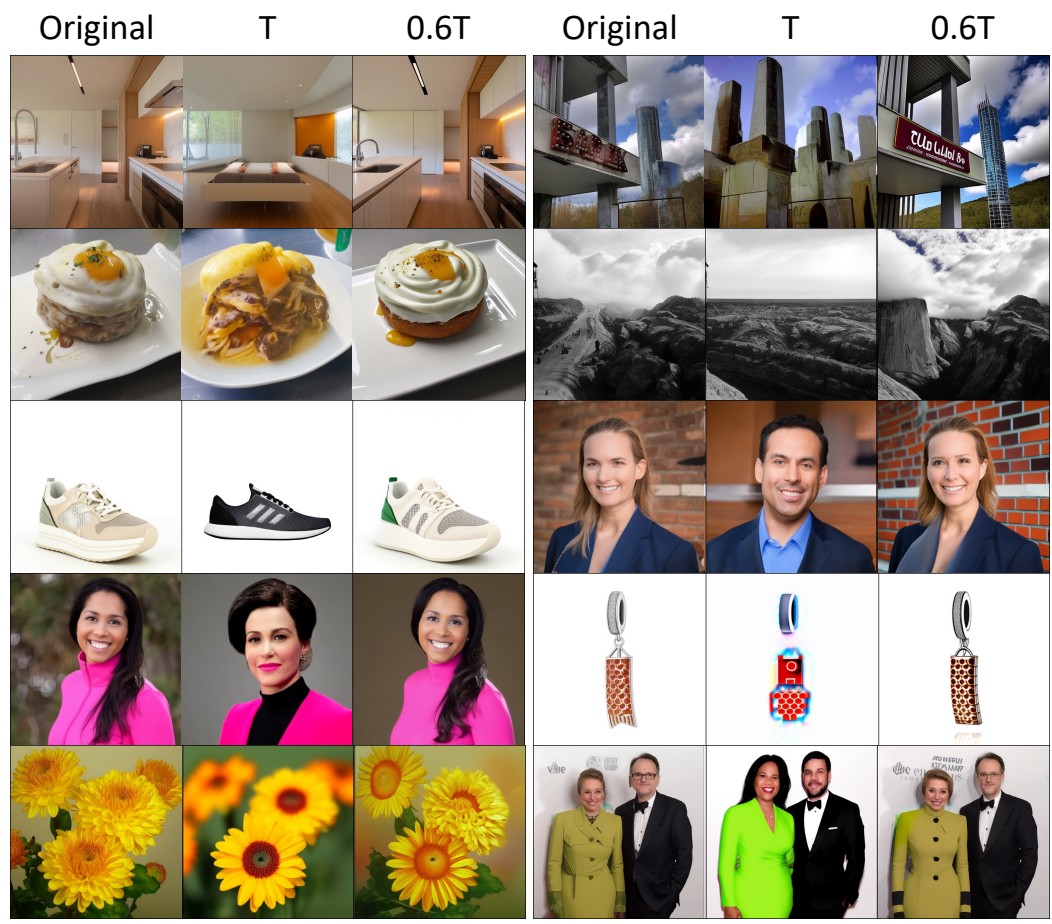

Figure A9: **More examples for Figure 5**.

