# OpenReview forum: "Learning Discrete Concepts in Latent Hierarchical Models"
_NeurIPS.cc/2024/Conference — NeurIPS 2024 poster_

### Official Review · Reviewer_BR1s · 2024-07-02

**Soundness:** 3
**Presentation:** 2
**Contribution:** 2
**Rating:** 5
**Confidence:** 2

**Summary:**

This paper studies the framework that identifies the discrete hierarchical latent variables for learning concepts from observed data examples. The proposed theory can be used to interpret the generating process of latent diffusion probabilistic models from the perspective of constructing object concepts.

**Strengths:**

1. The paper is, in general, well-written and well-motivated and focuses on the difficult problem of capturing concepts in vision problems.

 2. This work includes thorough theoretical derivation and details.

3. The illustrations of synthetic data demonstrate the applicability of the proposed method, and the results look interesting.

**Weaknesses:**

This paper is nice to read, while I have only limited experience in such a causal hierarchical modelling area. My questions can be found below.

1. From Sec. 2, given the description that the continuous latent variables c seem to control a lower level of features of data, while in Figure 1. a, it seems to be independent to concept factor d at the same level of the hierarchical structure. Could you elaborate on the relation and difference between c and d?

2. For LD experiments, at the early steps of the diffusion process (i.e., T), Figure 5 presents controlling of bread and species (high-level) features, while the low-level (e.g., object angle, background) can remain the same. But, in Figure A8, such capability does not hold, especially for the shoes. Can the author explain the reason behind this?

**Questions:**

Please see the weakness.

**Limitations:**

Yes.

---

> ### Author Rebuttal · Authors · 2024-08-07
>
> Thank you for your thoughtful review and dedicating your valuable time to our work! We address your concerns as follows.
>
>
> >W1: “From Sec. 2, given the description that the continuous latent variables c seem to control a lower level of features of data, while in Figure 1. a, it seems to be independent to concept factor d at the same level of the hierarchical structure. Could you elaborate on the relation and difference between c and d?”
>
> Sure, we are happy to elaborate! As you correctly observed in Figure 1.a, the continuous variable $\mathbf{c}$ directly influences the observed variable (e.g., images) $\mathbf{x}$ and is independent of the discrete variables $\mathbf{d}$ (please also see Equation 1).
>
> Intuitively, $\mathbf{d}$ contains all the discrete concepts/information in the image distribution, for instance, object classes or categories of shapes, whereas $\mathbf{c}$ captures the information, such as illumination, angles. These two sources of information are complementary and together fully describe the image information.
>
> Please note that the independence is not restrictive because even if $ \mathbf{c} $ depends on $ \mathbf{d} $ we can reduce it to the independent case by replacing $ \mathbf{c} $ with its the exogenous variable, which is independent of $\mathbf{d}$.
>
> Please let us know if we have cleared your question – thank you!
>
>
> >W2: “For LD experiments, at the early steps of the diffusion process (i.e., T), Figure 5 presents controlling of bread and species (high-level) features, while the low-level (e.g., object angle, background) can remain the same. But, in Figure A8, such capability does not hold, especially for the shoes. Can the author explain the reason behind this?”
>
> Thank you for the interesting question! We consider the object angle as a continuous variable (i.e., part of $\mathbf{c}$ in Figure 1.a) due to its continuous nature, so it does not belong to the discrete hierarchical structure. We speculate the consistent camera angles in Figure 5 result from the high probability of this specific angle in the data distribution, since most of the close-up face photos are captured from this angle in the training dataset.
>
> ---
> Please let us know if you have remaining issues -- we are more than happy to engage!

---

> > ### Author Response · Authors · 2024-08-12
> >
> > Dear Reviewer BR1s,
> >
> > As the rebuttal deadline approaches, we are wondering whether our responses have properly addressed your concerns? Your feedback would be extremely helpful to us. If you have further comments or questions, we hope for the opportunity to respond to them.
> >
> > Many thanks,
> >
> > 7636 Authors

---

> > > ### Comment · Reviewer_BR1s · 2024-08-12
> > >
> > > Thank the author for the response, and I am sorry for the late reply. I read through the comments, and most of my concerns are addressed, so I tend to keep my accept score.

---

> > > > ### Author Response · Authors · 2024-08-12
> > > >
> > > > Thank you so much for your valuable feedback! Since we have cleared your concerns, we were wondering whether it would be appropriate to update your recommendation from borderline accept to weak accept?

---

### Official Review · Reviewer_ae89 · 2024-07-10

**Soundness:** 3
**Presentation:** 2
**Contribution:** 3
**Rating:** 5
**Confidence:** 4

**Summary:**

This work introduces a novel identifiability analysis for a hierarchical latent models where latent variables are discrete and observations are continuous. The novelty of the result resides in the fact that previous results consider mainly continuous latent variables or make stronger assumptions on the form of the latent graph. An algorithm based on the theory is proposed and tested on synthetic data. Analogies between the approach and diffusion models are drawn.

**Review summary**
Overall I believe the theoretical contribution is interesting, important and novel, but the presentation requires some non-trivial restructuring since at the moment, a significant portion of the content of the paper is relayed to the appendix which makes it very hard to follow. I also thought the section on Diffusion Models was unconvincing and a bit disconnected from the rest of the contributions. I provided some suggestions, including submitting to a venue that allows for more space, like JMLR for instance. Given this, I can only recommend borderline acceptance.

**Strengths:**

- I believe the problem of identifiability in hierarchical latent variable models is interesting and important.
- The theory presented seems non-trivial and valuable (I did not read the appendix)
- Most identifiability results assumes continuous latent variables, so I was pleased to see further progress made in the case of discrete latents, which is much less common in the literature.
- I appreciated the high-level explanation of the proof technique between lines 223-233 which makes the connection to prior work transparent.
- The work is transparent about its limitations.
- Many examples are presented, which is helpful to understand the complex notions.

**Weaknesses:**

**Writing**

I thought the writing was quite good and easily understandable up until Section 3.3, where quality started degrading in my opinion. It really looks like the authors were running out of space and decided to relay *a very large* portion of the content to the appendix. Here's a (probably non-exhaustive) list of important concept and contributions which were relayed to the appendix:
- t-separation
- non-negative rank
- the minimal-graph operator
- the skeleton operator
- Condition A3.15
- Algorithm 1 (this is the main practical contribution!)
- adaptive sparsity selection mechanism for capturing concepts at different levels (another practical contribution)
- The literature review.

I can understand when a proof or even when a few very technical assumptions are kept in the appendix, as long as it does not interfere with understanding what is said in the main paper. But here, all these notions are referred to in definitions and assumptions and this really makes some sections unreadable. Also, some of these notions are not standard at all, like t-separation (I'm familiar with d-seperation) or non-negative rank (I'm familiar with the standard notion of rank) and would benefit from explanations in the main text.

In addition, Algorithm 1, which is the main practical contribution, is described only in the appendix. Same thing for the adaptive sparsity selection mechanism for capturing concepts at different levels in diffusion models from Section 6.2. The literature review is in the appendix.

**Diffusion models experiments**

I appreciate the effort to include more realistic experiments in a theoretical paper, but here I felt like Sections 5 & 6 on diffusion models were disconnected from the rest of the paper… My understanding is that the authors do not apply Algorithm 1 developed so far to the diffusion model. It seems the point of these sections is to draw what I believe to be very vague connections between the assumptions of their hierarchical model and the hierarchical nature of diffusion models. Section 6 only shows that different noise level of the latent space of a diffusion model correspond to our intuitive sense of “abstract levels”. But AFAIK this is a well known observation, no? Section 6.2 introduces another algorithm with only very high-level explanations with details in appendix.

**Suggestions for improvements**

I believe this manuscript would be more suited for a journal like JMLR than for a conference. The additional space would allow the authors to present all definitions in the main text and give intuitions for their meaning (for instance, the definition of atomic cover is very dense and could benefit from more explanations and intuitions. The recursive nature of the definition makes it quite challenging to grasp IMO). This also avoid the endless back and forth between main text and appendix.

Another possibility would be to remove the section on latent diffusion models, but even then this might not be enough.

**Relatively minor points:**

- Line 130-132: I believe the estimators d_hat, c_hat, g_hat and \Gamma_hat should be defined more explicitly, given how crucial they are to the results. In this phrasing, it is not clear whether these are estimated on a finite dataset or the full population.
- Table 1 and 2 are not referred to in the main text.
- Condition 3.1: The notion of splitting a latent variable is not properly explained.
- Condition 3.3: By definition, the support of a random variable is closed. See for example: https://en.wikipedia.org/wiki/Support_(mathematics)#In_probability_and_measure_theory . The only subsets of Rn that are both open and closed are the empty set and Rn itself. I’m guessing the authors were hoping to include more sets in their theory. It might be possible by assuming the set is “regular closed”, which means it is equal to the closure of its interior. This was done in a similar setting in [65].
Interesting to see that (iii) resembles the notion of G-preservation from [a] (see Definitions 11-12 and Proposition 3)
- Be careful with phrasing like line 237 “We define t-separation in Definition A3.2” as it sounds a bit like the authors are introducing this notion, but it’s not the case (source is cited properly in appendix).
- Confusion around t-separation: In Theorem 3.5, it is written “L t-separates A and B in G”, but the definition of t-sep refers to a tuple, i.e. “(L_1, L_2) t-separates A and B in G”. Not sure what the statement means.
- Text is too small in Figure 3
- Line 119: The definition of pure child was a bit confusing. In particular I thought B could contain more nodes than just the parents of A. Why not just repeat Definition A3.8 in the main text? (ne need to have a definition environment)
- Typo on line 141, V_1 or v_1 ?

**Questions:**

Condition 3.1
- The full support condition seems a bit strong, can the author discuss what it would mean for the running example with the dog?
- The function ne(v) was not defined, this is neighbors of v, right? It’s the union of Parents and children of v, correct?

The sparsity condition of Condition 3.3(iii) seems to be crucial for disentanglement in Theorem 3.4. How does this assumption compare to other works using sparsity of the decoder for disentanglement, such as [b,c,d]? I really believe there should be a discussion comparing the graphical assumptions with those of [d].

Definition 3.6: At line 256, what is the support of a set of atomic covers? (Supp(C) ?)

**References**

[65] Sébastien Lachapelle, Divyat Mahajan, Ioannis Mitliagkas, and Simon Lacoste-Julien. Additive decoders for latent variables identification and cartesian-product extrapolation. Advances in Neural Information Processing Systems, 36, 2024.

[a] S. Lachapelle, P. R. Lopez, Y. Sharma, K. Everett, R. L. Priol, A. Lacoste, and S. Lacoste-Julien. Nonparametric partial disentanglement via mechanism sparsity: Sparse actions, interventions and sparse temporal dependencies, 2024.

[b] J. Brady, R. S. Zimmermann, Y. Sharma, B. Scholkopf, J. von Kugelgen, and W. Brendel. Provably ¨ learning object-centric representations. In Proceedings of the 40th International Conference on Machine Learning, 2023.

[c] G. Elyse Moran, D. Sridhar, Y. Wang, and D. Blei. Identifiable deep generative models via sparse decoding. Transactions on Machine Learning Research, 2022.

[d] Y. Zheng, I. Ng, and K. Zhang. On the identifiability of nonlinear ICA: Sparsity and beyond. In Advances in Neural Information Processing Systems, 2022

**Limitations:**

Limitations were discussed properly throughout the paper.

---

> ### Author Rebuttal · Authors · 2024-08-07
>
> Thank you for recognizing our theoretical contribution as “interesting, important, and novel” and we highly appreciate your detailed, constructive suggestions on the writing.
> In light of your suggestion, we have put a lot of effort into revising the paper to explain all the involved definitions clearly, as we will detail below.
> We hope that this contribution should be visible to the community sooner. Therefore, we’d like to follow the tradition of publishing the paper at NeurIPS.
> We’d appreciate your feedback – thank you in advance!
>
> >W1: Writing suggestions.
>
> We are grateful for your thoughtful suggestions. In light of your feedback, we have made the following modifications to improve the readability of Sec. 3.3:
> 1. Trek-separation and non-negative ranks: we have moved definitions in the appendix to Sec. 3.3 (original lines 234) and explained their distinctions with d-separation and ranks.
> 2. Minimal-graph operator: we have added references to Figure A1 (a)(b) in the main text and moved its definition to the original line 288.
> 3. Condition A3.15 and the skeleton operator: we have moved Theorem 3.9 to the appendix and keep the reference in lines 300-30. Consequently, we keep Condition A3.15 and the skeleton operator at the current location in the appendix.
> 4. Algorithm 1: as our theory only involves simple changes of the original algorithms in [20], we tentatively leave the algorithm at its original location. Instead, we have added the following line in line 299 to highlight the modifications “our modifications consist of changing the original rank to the non-negative rank over probability tables and setting discovered covers as discrete variables as mentioned above”.
> 5. Adaptive sparsity: we have moved its description (original lines 1013-1015) to Sec. 6.2.
> 6. Literature review: we feel that the introduction has covered key literature, so for now we leave it in the appendix and will move it to the main text if given one additional page for the final version.
>
> To compensate for the space, we have moved large chunks of discussion in Sec. 5 to the appendix, along with Theorem 3.9 and its discussion. We’d like to note that all these edits are limited to Sec. 3.3 and Sec. 5 locally and the global structure of the paper remains.
>
> We’d love to hear your feedback!
>
> >W2: “Diffusion model experiments.”
>
> Thank you for the comment. In light of your comments, a portion of Sec. 5 has been moved to the appendix, while the main paper highlights its connection to our main results and findings. In our humble opinion, the hierarchical model is a valuable framework for reasoning about diffusion generation processes. We believe this could provide inspiration for the community to advance towards more controllable and interpretable generative models.
>
> >W3.1: Estimator definitions.
>
> Many thanks! We have added the following to line 132: “where we assume access to the full population $p(\mathbf{x})$”.
>
> >W3.2: Table 1, 2 references,
>
> Thanks for the reminder! We’ve added references in lines 314, and 318 respectively.
>
> >W3.3: notion of splitting variables.
>
> Thanks a lot! We’ve replaced “splitting” as “(i.e., turning a latent variable $z_ {i}$ into $\tilde{z} _{i,1}$ and $\tilde{z} _{i,2}$ with identical neighbors and matched cardinalities $ |\Omega^{z} _{i} | =  | \tilde{\Omega}^{z} _{i,1} | + | \tilde{\Omega}^{z} _{i,2} | $ ).”
>
> >W3.4: Random variable support.
>
> Great point! We have modified “open” to “closed”, as this doesn’t affect our main proof.
>
> >W3.5: definition phrasing.
>
> Thanks! We’ve edited it to “We introduce … [43].”.
>
> >W3.5: t-separation notations.
>
> Thanks! We’ve edited it as “ a partition $(\mathbf{L} _{1}, \mathbf{L} _{2})$ t-separates…” in Theorem 3.5.
>
> >W3.6: small text.
>
> Thanks for pointing it out! We have updated the fonts.
>
> >W3.7: pure children clarification.
>
> You’re totally right! We’ve moved Definition A3.8 to line 119.
>
> >W3.8: typos.
>
> Thanks! We have corrected it to $v_{1}$.
>
> >Q1: Condition 3.1.
>
> Great question. We acknowledge that this condition may be strong and we wished to avoid it. Unfortunately, it seems technically necessary without additional assumptions. For instance, its necessity is discussed in [24] for one-layer mixture models and we believe this is also the case for hierarchical models. Since science is established step by step, we hope our results could serve as the basis for more relaxed conditions in the community.
>
> In the distribution of dog images, suppose “head” has only two children “eyes” and “nose”. This condition means for each type of “head”, all combinations of shapes of “eyes” and “nose” should have non-zero probabilities. In reality, some combinations may be rare but can still appear at extremely small probabilities. That said, we agree that there are certainly cases where this can be violated, e.g., deterministic relations.
>
> Yes, you’re totally right about $\text{ne}(v)$. We’ve included “$\text{ne}(v):= \text{Pa}(v) \cup \text{Ch}(v)$” in line 119 - thanks!
>
> >Q2: Discussion on sparsity conditions.
>
> Great suggestion! We’ve added the following discussion to the original line 184:
> “Condition 3.3-iii is related to the notation of sparsity in disentanglement literature. Brady et al. [b] divide the latent representation into blocks and assume no shared children among blocks, which can be stringent if one aims to identify fine blocks. Moran [c] assumes pure observed children for each discrete variable, which is strictly stronger than Condition 3.3-iii. The structural sparsity in Zheng et al. [d] implies Condition 3.3-iii. By contrapositive, if latent variable $z_{0}$’s children form a subset of a distinct variable $z_{1}$’s children, then we cannot find a subset of observed variables whose parent is $z_{0}$ alone.”
>
> >Q3: Definition 3.6 clarification.
>
> It’s the collection of all states of variables in the cover. We’ve defined this notation in line 236 – thanks!
>
> ---
>
> Please let us know if you have further concerns and we are more than happy to discuss more!

---

> > ### Author Response · Authors · 2024-08-12
> >
> > Dear Reviewer ae89,
> >
> > As the rebuttal deadline approaches, we are wondering whether our responses have properly addressed your concerns? Your feedback would be extremely helpful to us. If you have further comments or questions, we hope for the opportunity to respond to them.
> >
> > Many thanks,
> >
> > 7636 Authors

---

> > > ### Comment · Reviewer_ae89 · 2024-08-13
> > >
> > > I thank the authors for their diligent answer. I'm sure the changes mentioned in the rebuttal are going to improve the manuscript quite a bit. I'm going to keep my score as is.

---

> ### Author Response · Authors · 2024-08-14
>
> Thank you so much for your feedback. We're really glad to hear that you think our changes can improve the manuscript by quite a bit. Your suggestions have been incredibly helpful -- thank you again!

---

### Official Review · Reviewer_7auw · 2024-07-13

**Soundness:** 3
**Presentation:** 2
**Contribution:** 3
**Rating:** 6
**Confidence:** 3

**Summary:**

This paper introduces a theoretical framework for learning discrete concepts from high-dimensional data using latent hierarchical causal models. The key contributions are:

1) Formalizing concept learning as identifying discrete latent variables and their hierarchical causal structure from continuous observed data.
2) Providing identifiability conditions and proofs for recovering discrete latent variables and their hierarchical relationships.
3) Interpreting latent diffusion models through this hierarchical concept learning lens, with supporting empirical evidence.

The work bridges theoretical causal discovery with practical deep generative models, offering new perspectives on how concepts might be learned and represented.

**Strengths:**

1. Novel formalization of concept learning as a causal discovery problem, providing theoretical grounding for an important area of machine learning
2. Rigorous proofs for identifiability of discrete latent variables and hierarchical structures under relatively mild conditions
3. Flexible graphical conditions that allow for more complex hierarchical structures than previous work
4. Interesting connection drawn between the theoretical framework and latent diffusion models, with empirical support
5. Clear potential impact on understanding and improving deep generative models for concept learning

**Weaknesses:**

1. The identification conditions (Condition 3.3 and 3.7) may be too restrictive for real-world scenarios. For instance, the invertibility requirement on the generating function g (Condition 3.3-ii) could be difficult to guarantee in practice, especially for complex high-dimensional data.
2. The method relies heavily on rank tests of probability tables (Theorem 3.5), which can be computationally expensive and numerically unstable for large state spaces or when probabilities are close to zero.
3. The approach assumes a clear hierarchical structure among concepts, but real-world concepts often have complex, overlapping relationships that may not fit neatly into a DAG structure.
4. The theory doesn't address how to handle noise or uncertainty in the observed data, which could significantly impact the identification of discrete states and the overall graph structure.
5. While the connection to latent diffusion models is interesting, the paper doesn't provide a concrete mechanism to leverage the theoretical insights for improving diffusion model architectures or training procedures.

**Questions:**

1. How robust is the identification process to small violations of the invertibility condition (Condition 3.3-ii)? Are there relaxations of this condition that could make the method more applicable to real-world data while maintaining identifiability?
2. Your interpretation of latent diffusion models suggests a correspondence between diffusion steps and concept hierarchy levels. How might this insight be used to design a diffusion process that explicitly learns and respects a given hierarchical concept structure?
3. The theory assumes discrete latent variables, but the latent space in diffusion models is continuous. How do you reconcile this discrepancy, and could your framework be extended to handle continuous latent variables with discrete-like behavior?

**Limitations:**

The authors adequately discuss the limitations of their work.

---

> ### Author Rebuttal · Authors · 2024-08-07
>
> Thank you for your encouraging words and thoughtful comments! We address your concerns as follows.
>
> >W1: “The identification conditions (Condition 3.3 and 3.7) may be too restrictive… for complex high-dimensional data.”
>
> Thank you for your comments. Maybe counterintuitively, the high dimensionality of image data actually makes Conditions 3.3 and 3.7 more likely to hold.
>
> Specifically, for the invertibility condition, note that since $\mathbf{x}$ is generated by $\mathbf{z}$ ($\mathbf{x} := g(\mathbf{z})$), $\mathbf{x}$ cannot contain more information than $\mathbf{z}$. Thus, the non-invertibility problem arises only when $\mathbf{x}$ fails to preserve the information of $\mathbf{z}$. High dimensionality gives $\mathbf{x}$ sufficient capacity to contain such rich information and facilitates invertibility. This assumption is also well-accepted in the community on image data [32-34].
> This also applies to Condition 3.3-iii and Condition 3.7-ii: with a large dimension of the observed variable $\mathbf{x}$, the children of each latent variable $d$ are less likely to overlap (Condition 3.3-iii), and latent variables are more likely to have unique, observed descendants (Condition 3.7-ii).
>
> At the same time,  we acknowledge that there do exist situations where some assumptions are violated. However, since science is constructed step by step, we hope our results can shed light on further development of this field.
>
> >W2: “...rank tests (Theorem 3.5), which can be computationally expensive and numerically unstable…”
>
> Thank you for raising this concern. We agree that rank tests are not easy (as noted in lines 415-417). However, we hope and believe that as these tests become increasingly important [19,20,41,42], more stable and efficient methods will be developed shortly. Therefore, we hope the current limitations of rank tests do not diminish the value of this contribution.
>
> >W3: “... overlapping relationships that may not fit neatly into a DAG structure”.
>
> By "overlapping relationships," we were wondering if you referred to cyclic relations among latent variables. (Please let us know if we've misunderstood this.) We agree that this paper focuses on the DAG structure. If non-DAG is necessary to address specific issues, we believe the ideas articulated in this paper can still provide valuable insights, although many practical issues should be considered.
>
> >W4: “... how to handle noise or uncertainty in the observed data..”
>
> Thank you for raising the issue of noise or uncertainty in the observed data, which is a significant and challenging problem in causal discovery and causal representation learning, as noted in recent contributions [a]. Addressing these problems is highly nontrivial, so we believe it should and will be tackled after solutions to the basic settings are clear, which is what we aim to achieve in this paper.
>
> [a] Causal Discovery with Linear Non-Gaussian Models under Measurement Error: Structural Identifiability Results. Zhang et al. UAI 2018.
>
> >W5: Concrete mechanism for improving diffusion models.
>
> Thank you for the comment. In our initial attempts, we applied basic ideas like sparsity to improve concept extraction techniques from the diffusion model, as shown in our experiments (Section 6.2 & Figure A6). We are actively working on building more principled generative models using our theoretical insights. Additionally, we hope the connection to hierarchical causal models presented in this work can inspire the community to develop more controllable and interpretable generative models.
>
> >Q1: robustness to invertibility violation and possible relaxations.
>
> This is a great question. We have been working on this for a long time. Recently, it seems hopeful to solve this problem by greatly extending [b]. While they consider a single latent variable in the probabilistic setting, extending to multiple latent variables seems highly possible but requires substantial investigation.
>
> [b] Instrumental variable treatment of nonclassical measurement error models. Hu et al. Econometrica, 2008
>
> >Q2: “... to design a diffusion process that explicitly learns and respects a given hierarchical concept structure?”
>
> Great question! We believe this correspondence can benefit the controllability of current diffusion models, which remains challenging for existing methods [c,d]. Unlike standard diffusion models, our framework suggests injecting different concepts at various diffusion steps, ensuring that concepts at different levels are properly rendered in the image [d]. We see more structured language and image interaction as a promising direction where our work could provide valuable insights.
>
> [c] Compositional Text-to-Image Generation with Dense Blob Representations. Nie et al. ICML 2024.
>
> [d] ELLA: Equip Diffusion Models with LLM for Enhanced Semantic Alignment. Hu et al. ArXiv 2024.
>
> Q3: The discrepancy between discrete variables and continuous diffusion latent space; possible extensions to continuous latent variables.
>
> Thank you for the question. We view the continuous latent space of diffusion models as an ensemble of embedding vectors for discrete concepts, similar to word embeddings in NLP (see lines 330-335). Park et al. [52] and our experiments (lines 388-399) support this interpretation, showing that the continuous space can be decomposed into a finite set of basis vectors, each carrying distinct semantic information.
>
> Regarding potential extensions, our theoretical analysis is connected to identification results on continuous latent variable models [19,20] (see lines 223-233) and can be adapted accordingly. We should note that for continuous models, strong assumptions like linearity are still required [19,20]. We hope to develop a completely non-parametric framework to handle such models in the near future.
>
> ---
>
> Please let us know if you have further questions -- thank you so much!

---

> > ### Author Response · Authors · 2024-08-12
> >
> > Dear Reviewer 7auw,
> >
> > As the rebuttal deadline approaches, we are wondering whether our responses have properly addressed your concerns? Your feedback would be extremely helpful to us. If you have further comments or questions, we hope for the opportunity to respond to them.
> >
> > Many thanks,
> >
> > 7636 Authors

---

> > > ### Comment · Reviewer_7auw · 2024-08-12
> > >
> > > I have no further concerns and have adjusted my score accordingly.

---

> > > > ### Author Response · Authors · 2024-08-12
> > > >
> > > > We are happy to see that we've resolved your concerns. Thank you so much for your dedicated time and your insightful reviews!

---

### Official Review · Reviewer_ZL2x · 2024-07-13

**Soundness:** 3
**Presentation:** 3
**Contribution:** 2
**Rating:** 7
**Confidence:** 1

**Summary:**

This paper presents a theoretical framework for learning discrete concepts from high-dimensional data using latent hierarchical models.

The authors propose formalizing concepts as discrete latent causal variables within a hierarchical causal model, and discuss under which condition the identification of these concepts and their relationships is possible. The theoretical contributions identifying those conditions and providing both theoretical insights and empirical validation using synthetic data, along with an interpretation of latent diffusion models through the proposed framework.

**Strengths:**

First, I apologize to the authors as I am not at all an expert in causal inference, and highly unsure about my remark (whether there be positive or negative), I also would like to mention to the author that I ensure myself the AC is aware of this.

Nevertheless, concerning the strength that I identified:

1. Up to my knowledge, the paper introduces an interesting formalization of what diffusion model are doing: learning concepts over hierarchical models, which is an interesting viewpoint and could help us better understand those models.
2. The identification conditions and theorems seems well-formulated (again, not an expert).
3. The authors try to validate their theoretical claims with real data experiments, I especially like the 6.1 which seems to partially verify their claim

**Weaknesses:**

Nevertheless, according to me (again, not an expert) this paper has problems, some more important than others.

So I will separate them into major problems (**M**) and minor problems (_m_). I want to make it clear that for me, all these problems are solvable and do not detract from the quality of the paper.

Let's start with what I think are the Major problems (**M**):

**M1**. Practicality of Recovering Hierarchical Graphs:
- I am left wanting more; the theoretical framework seems solid, but I would like to see concrete results. For example, can you recover the concept tree for the dog class in Stable Diffusion (or smaller diffusion model) ? Demonstrating this would undeniably highlight the paper's value and lead to a clear acceptance from my side. As it stands, I wonder if this framework could eventually teach us anything about diffusion models.


**M2**. Empirical Evidence for Real-world Data:
- The paper partially validates its claims using synthetic data. Even Section 6.1 is great, but clearly not enough to validate your claim. I'd say Figure A.4 in the appendix is another proof, and I expected maybe a comparison of this sparsity level with a real hierarchical model. Could we recover any information from the tree using this sparsity curve?

**M3**. Realism of Condition 3.3:
- There are doubts about the practicality and realism of Condition 3.3, especially 3.3-ii. It would be beneficial to discuss how realistic these conditions are in practical scenarios and provide more context or examples to substantiate them.

Now for the minor problems:

_m1_. Partial Literature Review:
- The related work section doesn't discuss concept extraction, which is a significant field with many papers every year at this conference. To me, you should at least cite or mention this literature.

_m2_. Clarity of Theoretical Explanations:
- Some parts of the theoretical explanations, particularly in Sections 3.2 and 3.3, are dense and may be difficult for readers to follow. Additional clarifications and examples would improve the accessibility of these sections.

**Questions:**

- Given my limited expertise, I am curious about the applicability of this framework in a supervised setting, particularly in relation to classification tasks. Could this framework be adapted, to show for example that supervised model learn only a part of the hierarchical graph relevant to classification problems?

**Limitations:**

Yes, the limitations identified by the authors are accurate and well-documented. Regarding the weakness I mentioned, I reserve the right to increase the score if the authors adequately address my major concerns.

---

> ### Author Rebuttal · Authors · 2024-08-07
>
> Thank you for your careful assessment and thoughtful comments on our work! We address your concerns point to point as follows.
>
> >M1. "Practicality of Recovering Hierarchical Graphs".
>
> Thank you for the great question! In light of your suggestion, we’ve included in our revision the following experiment, where we extract concepts and their relationships from Stable Diffusion through our hierarchical model interpretation. Please find the results in the submitted PDF file in the global response.
>
> Our recovery involves two stages: determining the concept level and identifying causal links. We add a textual concept, like "dog", into the prompt and identify the latest diffusion step that would render this concept properly. If "dog" appears in the image only when added at step 0 and "eyes" appears when added from step 5, it indicates that "dog" is a higher-level concept than "eyes". After determining the levels of concepts, we intervene in a high-level concept and observe changes in low-level ones. No significant changes indicate no direct causal relationship.
>
> As shown in the submitted PDF, we explore the relationships among the concepts "dog", "tree", "eyes", "ears", "branch", and "leaf". We provide the final recovered graph and some intermediate results of both stages. Although the current experiment scale (number of involved concepts) is relatively small, we hope these results demonstrate how to leverage the hierarchical causal model in our work to investigate black-box diffusion models.
>
> >M2: "Empirical Evidence for Real-world Data".
>
> Thank you for the interesting question. Given your question, we spent quite some time attempting to develop a quantitative relation between the attention sparsity and the graphical structure. Unfortunately, the gap seems nontrivial given that these two quantities belong to quite different objects (causal graphs, deep-learning layers) -- the same attention sparsity may correspond to a large family of causal graphs. For now, we think this metric may better serve as a qualitative indicator of the graphical connectivity, as in the manuscript.
>
> >M3: "Realism of Condition 3.3".
>
> Thank you for pointing this out. Given your suggestions, we’ve included the following discussion in our revised manuscript (original line 184).
>
> Condition 3.3 i: “In practice, the continuous variable $\mathbf{c}$ often controls extents/degrees of specific attributes (e.g., sizes, lighting, and angles) and takes values from intervals (which are connected spaces). For instance, the variable related to “lightning” ranges from the lowest to the highest intensity continuously.”
>
> Condition 3.3 ii: “For images, the invertibility condition assumes that images preserve the semantic information from the latent variables. Thanks to their high dimensionality, images often have adequate capacity to contain rich information to meet this condition. For instance, the image of a dog contains a detailed description of the dog’s breed, shape, color, lighting intensity, and angles, all of which are decodable from the image.”
>
> Condition 3.3 iii: “Practically, this condition indicates that lowest-level concepts influence diverse parts of the image. Lowest-level concepts are often atomic such as a dog’s ear, eyes, or even finder. As we can see, these atomic features often don’t overlap in the image (e.g., ears and eyes are separate).”
>
> >m1: "Partial Literature Review".
>
> Thank you for the helpful suggestion! Thanks to your pointer, we have included the following paragraph in our revision. Please let us know if we have overlooked important works and we would be happy to include them.
>
> “A plethora of work has been dedicated to extracting interpretable concepts from high-dimensional data such as images. Concept-bottleneck [12] first predicts a set of human-annotated concepts as an intermediate stage and then predicts the task labels from these intermediate concepts. This paradigm has attracted a large amount of follow-up work [13][a-e]. A recent surge of pre-trained multimodal models (e.g., CLIP [17]) can explain the image concepts through text directly [14-16]. In contrast with these successes, our work focuses on the formulation of concept learning and theoretical guarantees.”
>
> [a] Post-hoc Concept Bottleneck Models. Yuksekgonul et al. ICLR 2023.
>
> [b] Probabilistic Concept Bottleneck Models. Kim et al. ICML 2023.
>
> [c] Addressing Leakage in Concept Bottleneck Models. Havasi et al. NeurIPS 2022.
>
> [d] Incremental Residual Concept Bottleneck Models. Shang et al. CVPR 2024.
>
> [e] Interactive concept bottleneck models. Chauhan et al. AAAI 2023.
>
> >m2: "Clarity of Theoretical Explanations".
>
> Thank you for your suggestion! Besides the examples for Condition 3.3 as quoted above, we have included the following for Condition 3.7.
>
> "Intuitively, Condition 3.7-ii requires that each discrete variable has sufficiently many children and neighbors to preserve its influence while avoiding problematic triangle structures to ensure the uniqueness of its influence. Condition 3.7-iii requires sufficient side information (large $|\mathbf{A}|$) to identify collider $\mathbf{C}$."
>
> >Q: "applicability of this framework in a supervised setting".
>
> Thank you for the intriguing question! We believe this is possible. In the supervised setting, the prediction target is often a high-level concept. To make correct predictions, the model may need to leverage low-level concepts, like ears and fur for classifying cats. Thus, one may impose sparsity constraints on the representation to view which features are engaged for predicting a specific class and infer certain concept structure therein. For example, if predicting cats and dogs calls for latent variables $(z_{1}, z_{2})$ and  $(z_{2}, z_{3})$ respectively, one may infer that "cat", as a high-level variable, is a parent to $ (z_{1}, z_{2}) $, and "dog" is a parent to $ (z_{2}, z_{3}) $.
>
> ---
>
> Please let us know if it is unclear and we would be happy to discuss further!

---

> > ### Author Response · Authors · 2024-08-12
> >
> > Dear Reviewer ZL2x,
> >
> > As the rebuttal deadline approaches, we are wondering whether our responses have properly addressed your concerns? Your feedback would be extremely helpful to us. If you have further comments or questions, we hope for the opportunity to respond to them.
> >
> > Many thanks,
> >
> > 7636 Authors

---

> > ### Comment · Reviewer_ZL2x · 2024-08-12
> >
> > Thank you for the detailed and thoughtful responses to my comments.
> >
> > I appreciate the additional experiments and explanations you've provided, especially the practical applications (M1, M2) of your hierarchical model interpretation and the discussion of Condition 3.3.
> >
> > For m1, great, it's excellent, I was also thinking about concept extraction (ACE, CRAFT, ICE) which I think share some motivation with your work.
> >
> > Overall, I want to mention that your work has given me a lot to think about lately, and I find it very intriguing. Thank you. Given this, I’m raising my score to 7. I am still not an expert in causal inference, but I believe this work is interesting, especially for the XAI community.
> >
> >
> > Good luck with the acceptance, and thank you again for this work!

---

> > > ### Author Response · Authors · 2024-08-12
> > >
> > > Thank you so much for your kind and encouraging words – we are truly grateful for your positive and constructive feedback! It is rewarding to know that our research has provided you with new ideas to consider and we do hope our work will contribute meaningfully to the XAI community.
> > >
> > > Thank you once again for your thoughtful review and best wishes, and we will continue to explore this exciting direction!

---

### Author Rebuttal · Authors · 2024-08-07

We sincerely thank the reviewers for their efforts and helpful comments regarding our paper. We are encouraged that all reviewers appreciate our theoretical contribution to the identifiability of latent discrete hierarchical graphs and our novel formalization of concept learning as a latent causal discovery problem. These contributions provide an intriguing perspective on latent diffusion models and have the potential to enhance the understanding and improvement of these models. Reviewer BR1s also highlights that the paper is well-written and enjoyable to read, while Reviewer ae89 values the provided examples that aid in comprehending the complex concepts.

Below, we give a summary of the responses:
* To Reviewer ZL2x: we’ve included additional experiments to extract concept structures from Stable Diffusion and attached the PDF file here. We’ve also included an additional literature review as suggested by the reviewer and clarification/interpretation of our theoretical conditions.
* To Reviewer 7auw: we’ve responded to our theoretical conditions, rank tests, graphical structures, and robustness. We’ve shared the potential usage of our work to advance the field of diffusion models.
* To Reviewer ae89: we’ve re-organized Sec. 3.3 and Sec. 5 to improve the readability of our work. We’ve corrected typos, added the references of Tables 1 and 2, and included a discussion of sparsity conditions.
* To Reviewer BR1s: we’ve elaborated on the formulation and experimental results.

Please see our detailed responses to each reviewer. We hope our revisions and explanations satisfactorily address the concerns raised. Once again, thank you for generously contributing your time and expertise to the community.

---

### Decision · Program_Chairs · 2024-09-25

**Decision:**

Accept (poster)

**Comment:**

Scores 5 5 6 7

Great paper.  Well written and includes meaningful theoretical derivations and good experiments on synthetic data.

The reviewers were quite positive about the paper, some even enthusiastic.  One criticism was (reviewer ae89) that too much content is contained in the appendix.  However, in the rebuttal the authors pointed out how they restructured the paper, they added further experiments and added some discussion about how this relates to diffusion models.

So, this paper is a clear accept for me.